# Rainfall patterns in the 24 East Asian mega-cities using a complex network

Kyunghun Kim[1], Jaewon Jung[2], Hung Soo Kim[1], Masahiko Haraguchi[3], and Soojun Kim[1]

[1]Department of Civil engineering, INHA University, Incheon, 22212, Rep. of Korea
[2]Department of Hydro Science and Engineering Research, Korea Institute of Civil Engineering and Building Technology, Gyeonggi-do, 10223, Republic of Korea
[3] Research Institute for Humanity and Nature, Kyoto, Japan

*Correspondence to*: Soojun Kim (sk325@inha.ac.kr)

**Abstract.** Concurrent floods in multiple locations pose systemic risks to the interconnected economy in East Asia through supply chains. Despite the significant economic impacts, the understanding of the interconnection between rainfall patterns in the region is yet limited. Here, we analyzed spatial dependence in rainfall patterns of the 24 mega-cities in the region using complex analysis theory and discussed the technique's applicability. Each city and rainfall similarity was represented by a node and a link, respectively. Vital node identification and clustering analysis were conducted using adjacency information entropy and multiresolution community detection. In the vital node identification analysis result, high-rank nodes are cities that are located near main vapor providers in East Asia. Through the multiresolution community detection, the groups were clustered to reflect the spatial characteristics of the climate. In addition, the climate links between each group were identified through the cross-mutual information considering the delay time for each group. We found a strong bond between northeast China and the south Indochina Peninsula and verified that the links between each group were originated from the summer climate characteristics of East Asia. The result of the study shows that complex network analysis could be a valuable method for analyzing the spatial relationship between climate factors.

Keywords: Concurrent floods, Complex network, Adjacency information entropy, Multiresolution community detection

## 1 Introduction

East Asia accounts for 54% of the global supply chain, providing a wide range of services and products across the world (Ann et al, 2020). However, East Asia is prone to major floods. According to disaster database of the Centre for Research on the Epidemiology of Disasters (CRED), which offer essential core data on the occurrence and effects of disasters all over the world, an annual average of 165 flood disasters occurred worldwide during the period from 2000 to 2020, resulting in 5,278 deaths and economic damage up to $29 million. While more than 22% of these flood disasters have occurred in East Asia, more than 60% of global-flood deaths and economic damage worldwide are in the region. For instance, Thailand recorded 813 deaths and $40 million worth of damages from floods in 2011 (Haraguchi and Lall, 2015), while China recorded 300 fatalities and $4.5 million in damages from floods in 2019 (CRED). These flood damages occurred in several areas of East Asia simultaneously. Even though floods occurred simultaneously at distant places, the impacts of floods will propagate through supply chains, incurring economic losses in the entire region. In this sense, concurrent flooding causes severe life and economic losses in multiple countries at the same time, disrupting the global economy more severely. For example, in 2020, concurrent floods in East Asia inundated automobile factories in Thailand, disrupting automobile supply, adversely affecting China's rare-earth and fertilizer industries along the Yangtze river, and affecting the global rare-earth industry (AON. 2020).

Changes in rainfall characteristics caused by climate change are some of the primary causes of concurrent floods in East Asia. These changes occur across all regions, and the changed characteristics affect each other, resulting in even more significant changes (Wang et al., 2020). Therefore, it is vital to investigate the relationships among rainfall patterns in each region. Many studies have been conducted to identify rainfall relations in East Asia. Most of them investigated the relationship between

major East Asian countries using statistical techniques (Jeong et al., 2008, Kosaka et al., 2011, Deng et al., 2014), and some demonstrated connections among weather factors, sea-level temperature, and monsoons. (Wu et al., 2003, Lau and Kim., 2006, Li et al., 2010, Sun and Wang, 2012, Renguan, 2017). Researchers have also used teleconnection methods to discover relationships between precipitation in East Asia and other parts of the world (Kripalani and Kulkarni, 2001, Sahai et al., 2003,

Riyu and Zhongda, 2009, Lin, 2014, Maity et al., 2020.). These studies have been used to anticipate rainfall in East Asia and aid in preparing for flood disasters. This study investigated the usefulness of complex network concepts for relationship analysis.

Complex network theory, developed by Leonard Eüler in 1735, expresses and analyzes a subject or phenomenon as a graph. In the late 1990s, Watts and Strogatz (1998) and Barabási and Albert (1999) extended the analytical technique, making the

theory fundamental in network science. A complex network can display a complicated phenomenon as a simple graph. Information obtained from the methodology can be used to identify the characteristics of subjects, their physical behavior, and the roles and relationships of the phenomenon's components. Complex network analysis has also been used in various fields because of its high applicability. For example, researchers have applied it to social networks (Michael et al., 2010), world trade (Bader et al., 2007), air transportation nets (Cardillo et al., 2013), patterns in human migration (Davis et al., 2013), and others.

The analytical method has also been used in the fields of hydrology and meteorology fields to discover new patterns and relationships (Donges et al., 2009, Scarsoglio et al., 2013, Boers et al., 2015, Joo et al., 2021, Wolf et al., 2020). About precipitation related researches, the method had been used to analyze extreme rainfall patterns around the world (Boers et al., 2019), track rainfall events caused by typhoons (Ozturk et al., 2018), and study the spatial connectivity of rainfall (Ihsan et al., 2018) to determine new information or characteristics.

With the encouraging results of previous rainfall-related studies, this study applied complex network theory to rainfall in East Asia to understand the relationships between the rainfall patterns in each region. The complex networks in hydrology and meteorology fields defined connectivity using statistical interdependence methods. Therefore, characteristics can be analyzed from the relationships. In addition, for clustering analysis, complex network-based methods consider the entire network, rather than the regions independently, unlike many other traditional methods. This feature results in a more accurate clustering (Long

and Liu et al., 2019). Despite this advantage, one of the challenges in complex network theory is to identify thresholds, which determines whether the links existed. While no perfect methodology exists to clearly address this challenge, new methodologies are constantly being proposed. In this study, we assumed that each region (node) is connected with all the other regions (nodes) in the network, and that each connection (link) has a similarity as a weight. By using the similarity measures as weights, it makes the network reflect relationships between regions. Using the similarity measure makes a network the

reflect relationship between region. The link weight is one of the most important input for constructing and analyzing a network. We assessed the effects of each region through centrality analysis and grouped the regions according to clustering analysis. Subsequently, Mutual Information (MI) was calculated with a time lag (i.e., cross-mutual information) to identify the relationships between each group. Past studies about complex network considered only spatial factors, while we add also temporal factors.

The remainder of this paper is organized as follows. Section 2 describes the study area and data used in this study. The complex network theory and related indicators are detailed in Section 3. Section 4 presents the results of the complex network analysis of East Asia and a discussion of these results. Section 5 presents the conclusions of this study.

## 2 Study area and materials

In this research, the major cities in East Asia are analyzed (Fig. 1). Among East Asia cities, we used selected cities selected

by Haraguchi et al. (2019). They chose the cities with more than 5 million people and a high degree of urbanization. Because, rainfall often causes numerous floods in the region, and with growing urbanization, more and more cities suffer from small

and medium-sized frequent floods, as well as large scale low-probability floods (The World Bank, 2015). However, we excluded Surabaya, Jakarta, and Badung (Indonesia) from the selected cities because of the changes in the location of rainfall observation since 2007. Instead, we included Ho Chi Minh City, Hai Phong (Vietnam), and Cebu (Philippines), which are economically emerging. Thus, a total of 24 cities were selected.


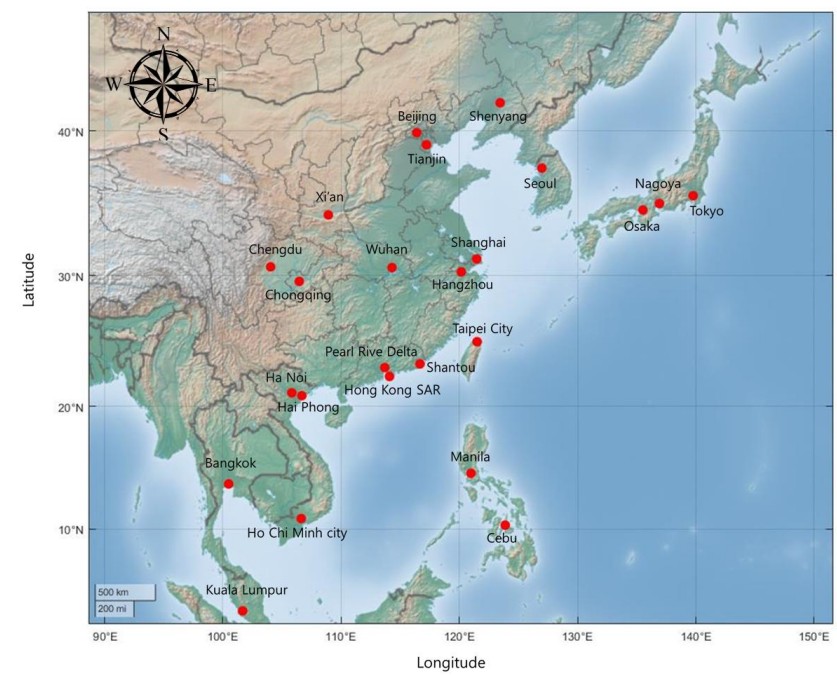

**Figure 1.** Selected 24 major cities in East Asia

This study used daily precipitation data from the Asian Precipitation – Highly Resolved Observational Data Integration Toward Evaluation (APHRODITE) grid precipitation dataset (Akiyo et al., 2012). The APHRODITE data contains long-term, high-resolution daily rainfall data of the Asian continent obtained from the dense precipitation observation data network (Fig.

2). The data were obtained from the APHRODITE Water Resource project conducted by the Research Institute for Humanity and Nature (RHIN) and the Meteorological Research Institute of Japan Meteorological Agency (MRI/JMA) and have been used in many studies because of their high definition. We extracted the rainfall data from the grid$(0.25° × 0.25°)$ where each city belongs.

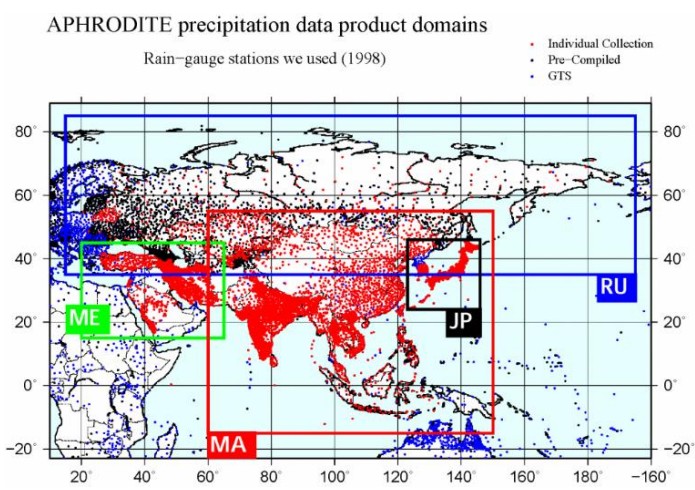


**Figure 2.** APHRODITE data (http://aphrodite.st.hirosaki-u.ac.jp/products.html ) of cities; Dots are example of station data distribution that were used for making data(Red: individual collections, Black: pre-complied data, Blue: global telecommunication networks (GTS) based data obtained from Global summary of day (NCEI/NOAA)). Rectangles show the domain of precipitation data. In this research, precipitation data on a 0.25 degree grid for Monsoon Asia (MA) for the period from 1981 to 2015.

Daily rainfall data for each city consisted of observations from January 1, 1981, to December 31, 2015. The basic statistics for each city's rainfall data are the same as those listed in Table 1.

**Table 1.** Basic statistics values for rainfall data of cities; Basic statistics contain average, standard deviation, coefficient of variation and skewness.

| Station | Average (mm/day) | Standard deviation | Coefficient of Variation | Skewness |
|---|---|---|---|---|
| Pearl River Delta | 4.277 | 9.416 | 2.202 | 4.063 |
| Tokyo | 3.637 | 10.736 | 2.952 | 7.044 |
| Shanghai | 2.985 | 6.446 | 2.160 | 3.993 |
| Beijing | 1.316 | 4.377 | 3.325 | 6.395 |
| Manila | 6.241 | 12.514 | 2.005 | 5.023 |
| Seoul | 3.457 | 9.427 | 2.727 | 5.543 |
| Osaka | 2.802 | 5.900 | 2.105 | 4.619 |
| Bangkok | 1.101 | 4.067 | 3.694 | 7.308 |
| Tianjin | 3.903 | 9.932 | 2.544 | 4.910 |
| Shantou | 2.211 | 4.853 | 2.195 | 5.279 |
| Chengdu | 3.782 | 6.241 | 1.650 | 3.246 |
| Ho Chi Minh City | 4.551 | 11.314 | 2.486 | 4.919 |
| Nagoya | 3.176 | 7.762 | 2.444 | 4.492 |
| Wuhan | 4.583 | 11.278 | 2.461 | 4.427 |
| Hong Kong SAR | 4.889 | 6.460 | 1.321 | 2.177 |
| Shenyang | 1.590 | 4.946 | 3.11 | 6.183 |
| Taipei | 6.955 | 16.128 | 2.319 | 6.192 |
| Hangzhou | 3.614 | 7.413 | 2.051 | 3.981 |
| Kuala Lumpur | 6.196 | 7.970 | 1.286 | 2.311 |
| Xi'an | 1.527 | 4.099 | 2.684 | 5.054 |
| Ha Noi | 4.053 | 8.870 | 2.189 | 4.417 |
| Chongqing | 2.790 | 5.751 | 2.061 | 4.890 |
| Cebu | 4.126 | 6.759 | 1.638 | 4.923 |
| Hai Phong | 3.801 | 9.530 | 2.507 | 5.023 |

Taipei records the largest rainfall amount on average, approximately six times more than that of Bangkok, which has the lowest. Bangkok has the largest variation, and Kula Lumpur has the least.

## 3 Methodologies

### 3.1 Complex network analysis

Complex network analysis effectively visualizes a subject or phenomenon using a network and analyzes its characteristics,
components, and relationships among nodes in the network. To apply complex network analysis, nodes and links must be defined. A node represents some entity or agent that serves as a point of intersection/junction within a network (Kivelä et al., 2014). For example, in the global airways network, airports become nodes. A link is an element that connects each node. In a

global airways network, airways are the links. Defining these two elements is crucial in the analysis because even networks with the same number of nodes and links can potentially makes various forms (Fig. 3).

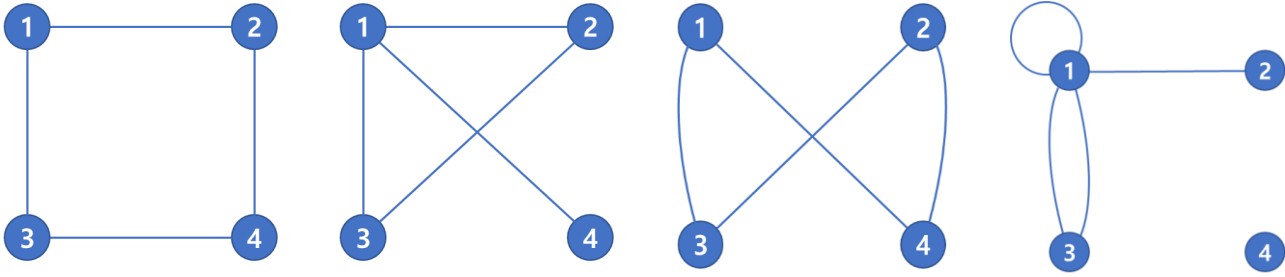

**Figure 3.** Various shapes of networks with the same number of nodes and links; Each network has 4 nodes and 4 links. However, those shows different shapes and have different topological characteristics.

In a complex network, links are the most influential aspects of the network. This is because the type and characteristics of the graph vary depending on the type of link used and how it is defined. Based on the directionality and weight of the link, the network can be an undirected/directed network or an unweighted/weighted network. Generally, it is easy to define links in transportation systems or power grid system, which show clear connections between elements. However, if uncertainty occurs in the connections like social networks, researchers must define them. The most widely used methodology is the similarity measure (Donges et al., 2009). Depending on the value of the similarity calculated between two nodes, the researcher can define whether a link exists. While various previous studies used the Pearson correlation coefficient for links, they tend to derive inaccurate values when they are applied to nonlinear data (Zadian et al., 2018). To address this problem, some researchers have utilized Mutual Information (MI) as an alternative (e.g., Donges et al., 2009, Kim et al., 2019, Ghorbani et al., 2021). MI is based on the information and probability theory. For two variables (A and B), it quantifies and represents the amount of information of B contained by the variable A.

$$\text{MI}(A,B) = \sum_{b \in B} \sum_{a \in A} p(a,b) log(\frac{p(a,b)}{p(a)p(b)}) \tag{1}$$

Here, $p(a)$ and $p(b)$ are probability distributions of variables. $p(a,b)$ indicates the joint probability density function of the variables. MI values range from 0 to $\infty$, and an MI of 0 indicates that the two variables are independent of each other. MI can consider the nonlinearity of the data and has the advantage of calculating the similarity between different data sizes (Goyal, 2014).

### 3.2 Vital node identification using adjacency information entropy

Important nodes in a network have various effects on the structure or function of the network. Identifying these nodes is of practical and theoretical value (Xu et al., 2020). For example, if a government identifies which places have a key role in power grids and traffic networks, it can effectively invest and create defense measures to prepare for blackouts and traffic jams. While some methods of identifying important nodes have been developed, these methods have limitations and are only applicable to certain types of networks (Mester et al., 2021) such as undirected network. Xu et al. (2020) developed a new methodology based on information entropy, making it applicable to all types of networks. This method has more efficient and accurate results than the existing methods. The procedure of the method is as follows (more detailed explanation about the Vital node identification method is in Xu et al. (2020)). The first step is calculating strength of each node in a weighted network (Eq. (2)).

$$k_i = \sum_{j \in \Gamma_i} w_{ji} \tag{2}$$

where, j is the neighbor of node i. $\Gamma_i$ is the set of neighbors of node i. $w_{ji}$ is weight of link that connect node j and node i. If a network is unweighted, degree is the number of neighbor nodes. Next, estimate an adjacency degree of each node (Eq. (3)).

$$A_i = \sum_{j \in \Gamma_i} k_j \tag{3}$$

$A_i$ is an adjacency degree which means total weight of neighbor nodes of node i. Based on Eq. (2) and Eq. (3), selection probability can be calculated (Eq. (4)). Eventually an adjacency information entropy of a node is calculated based on Eq. (5).

$$P_{i_j} = k_i / A_j \tag{4}$$

$$E_i = \sum_{j \in \Gamma_j} (P_{i_j} log_2 P_{i_j}) \tag{5}$$

After comparing the calculated adjacency information entropy of each node, the importance is determined according to the descending power. In the rainfall studies, vital nodes are interpreted as important points for propagation of rainfall event.

### 3.3 Multiresolution community detection in weighted complex networks

A complex network consists of many nodes and links. Some nodes with strong relationships or similar characteristics can be clustered together. These clusters have several features and perform specific network functions. However, the cluster results
depend on the level of analysis. Therefore, the multiresolution community detection method can be a useful method for understanding complex networks (Newman, 2012). Several cluster analysis methods have been used for complex networks, but they require intense computations for complicated network shapes and focus only on graphical properties (Long and Liu, 2019). To address these problems, Long and Liu (2019) proposed a new clustering methodology using an intensity-based community detection algorithm (ICDA) in weighted networks. This method has the advantages of forming groups more
accurately and faster than other methods. For making groups, belonging coefficient of the nodes should be calculated. First step for estimating belonging coefficient is defining a distinct path. The simple (not repeating links between nodes) and elementary (not repeating nodes) path θ between node i and j with k-edges is denoted as a k-edge distinct path, if the path has no identical intermediate nodes or edges with any other distinct paths. After defining distinct paths, it need to calculated link intensity of each link. The equation of link intensity is same as Eq. (6).

$$I_P(e_{ij}) = \begin{cases} \sum_{p=1}^{P} \alpha_p \times \dfrac{\sigma(path_p(v_i, v_j))}{\min(w_i, w_j)}, & e_{ij} \in E \\ 0, & otherwise \end{cases} \tag{6}$$

Where, $\sigma(path_p(v_i, v_j))$ is the sum of link weights in p-edge distinct paths from node i ($v_i$) to node j ($v_j$). P is the parameter of the path, and $\alpha_p$ is a polygonal effect parameter. For edge $e_{ij}$ between node i and node j, $w_i$ and $w_j$ are their respective strengths. Based on link intensity, find links which have larger value of link intensity than a threshold and create a group of nodes with the identified links (Eq. (7)).

$$v_j = \begin{cases} v_j \in V, & I_P(e_{ij}) > t \\ v_j \in c_u, & I_P(e_{ij}) > t \end{cases} \tag{7}$$

Where, t ($0 < t \leq 1$) is the selected threshold, and $c_u$ is a group of nodes. The threshold is determined according to
researcher personal view. Final step is calculating belonging coefficient ($I_P$) of the nodes in node set u ($c_u$) (Eq. (8)).

$$I_P(c_u, v_j) = \sum_{v_i \in c_u} I_P(e_{ij}) \tag{8}$$

For evaluating the result of the cluster method, we used Newman-Girvan modularity. The Newman-Girvan modularity method compares the number of links connecting nodes inside a group of nodes with an expectation of this number under a random null model (Newman & Girvan, 2004). The modularity (Q) is calculated by Eq. (9). Where, P is a cluster of node groups (P = $\{g_1, g_2, \cdots, g_a\}$), and $g_a$ is a node group. m is a total weight of links. The modularity measure assigns high scores
to communities if they are densely connected internally, but only weakly connected to other group. We set threshold groups divided by 2.5% intervals from 95% to 75% and calculated the Newman-Girvan modularity according to the threshold groups.

$$Q = \frac{1}{2m} \sum_{g_a \in P} \sum_{i,j \in g_a} (A_{ij} - \frac{k_i k_j}{2m}) \tag{9}$$

## 4 Application and Results

### 4.1 Construction of East Asia rainfall network

In this research, we design a rainfall network as a weighted and undirected graph. Each node was selected from 24 major cities, and link weights represented shared knowledge between nodes. Table 2 compares the results of the link weights of the nodes.

**Table 2.** Average, maximum, and minimum link weights of each node; Parentheses next to link weights are nodes that form a maximum or minimum value for target nodes;

| Node | Average | Maximum (Node) | Minimum (Node) |
|---|---|---|---|
| Pearl River Delta | 0.352 | 1.674 (Hong Kong SAR) | 0.203 (Tokyo) |
| Tokyo | 0.226 | 0.528 (Nagoya) | 0.140 (Tianjin) |
| Shanghai | 0.253 | 1.076 (Hangzhou) | 0.153 (Tokyo) |
| Beijing | 0.232 | 0.850 (Tianjin) | 0.130 (Osaka) |
| Manila | 0.294 | 0.467 (Ho Chi Minh City) | 0.219 (Shanghai) |
| Seoul | 0.227 | 0.276 (Ha Noi) | 0.155 (Tokyo) |
| Osaka | 0.244 | 0.870 (Nagoya) | 0.130 (Beijing) |
| Bangkok | 0.272 | 0.520 (Ho Chi Minh City) | 0.194 (Shanghai) |
| Tianjin | 0.252 | 0.850 (Beijing) | 0.143 (Osaka) |
| Shantou | 0.284 | 0.861 (Hong Kong | 0.183 (Beijing) |
| Chengdu | 0.293 | 0.621 (Chongqing) | 0.200 (Shanghai) |
| Ho Chi Minh City | 0.308 | 0.520 (Bangkok) | 0.217 (Shanghai) |
| Nagoya | 0.254 | 0.870 (Osaka) | 0.139 (Beijing) |
| Wuhan | 0.292 | 0.529 (Hangzhou) | 0.195 (Taipei City) |
| Hong Kong SAR | 0.364 | 1.674 (Pearl River Delta) | 0.215 (Tokyo) |
| Shenyang | 0.240 | 0.367 (Tianjin) | 0.166 (Tokyo) |
| Taipei | 0.220 | 0.333 (Shantou) | 0.147 (Beijing) |
| Hangzhou | 0.279 | 1.076 (Shanghai) | 0.164 (Beijing) |
| Kuala Lumpur | 0.308 | 0.377 (Wuhan) | 0.207 (Beijing) |
| Xi'an | 0.271 | 0.508 (Chengdu) | 0.188 (Osaka) |
| Ha Noi | 0.341 | 1.162 (Hai Phong) | 0.238 (Tokyo) |
| Chongqing | 0.289 | 0.621 (Chengdu) | 0.207 (Beijing) |
| Cebu | 0.246 | 0.354 (Manila) | 0.179 (Beijing) |
| Hai Phong | 0.342 | 1.162 (Ha Noi) | 0.235 (Shanghai) |

According to Table 2, the ranges of average, maximum, and minimum link weights are 0.22–0.37, 0.27–1.67, and 0.13–0.24, respectively. The standard deviation of average, minimum and maximum values are 0.041, 0.033 and 0.394. The standard deviation of maximum values had 10 times larger value than the minimum values. We observe that the cities with the maximum values for each node were closely located. This is because rainfall characteristics in cities located in near areas are similar; thus, the value of the MI is high. Each node has a maximum value with several different cities, while the minimum value is for certain cities such as Beijing and Tokyo. Beijing and Tokyo are selected as the cities with the lowest MI value eight and six times, respectively. Two cities have a common feature that their location is in the outskirt of the study area.

**4.2 Vital node identification by adjacency information entropy**

For the network, we apply Vital Node Identification (VNI) to determine the influence of nodes. VNI can be used to analyze all types of networks and more precisely determine the effects of nodes more accurately. The cities with high ranking nodes are located around the South China Sea (Fig. 4). In addition, they have a high adjacency in the adjacency matrix. Cities with low-rank nodes are in the northeast outskirts, except for Taipei, and have a low value of average MI. From these results, we can deduce that the location of a node affects its influence. However, location is not the only factor affecting vital node identification. Despite its proximity to the South China Sea, Taipei has a low rank because its link weight is the smallest on average (0.220). We also find another common thing between high-rank cities that they are located in the beginning of the impact of main influence factors in East Asia rainfall. We will describe this in Section 4.5.

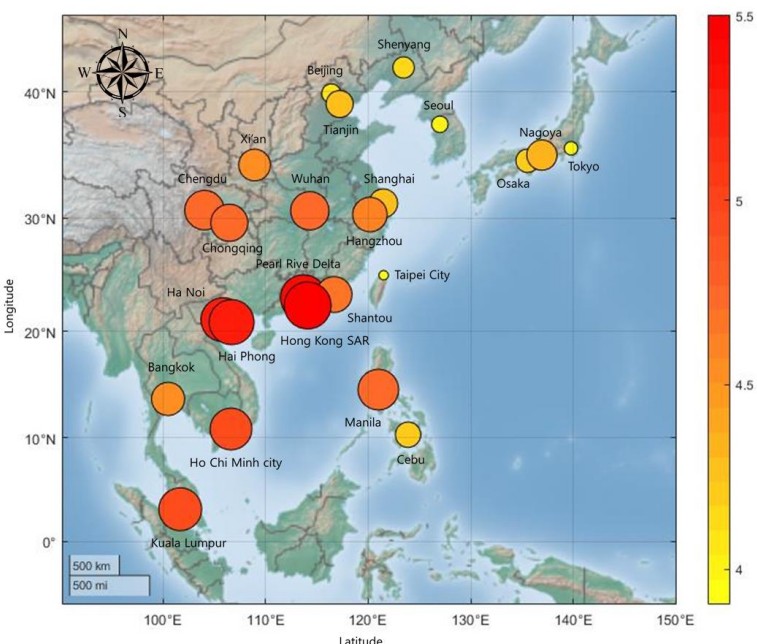

**Figure 4.** Adjacency information entropy value of cities; color and size of circle are respectively proportional to the entropy and rank. The right side of bar shows the adjacency information entropy values of nodes; except for Taipei city, nodes near the South China Sea, which had higher values;

**4.3 Clustering analysis using multiresolution community detection**

In clustering analysis, multiresolution community detection is applied to 24 nodes to create groups. After calculating the belonging coefficient, we determine the groups based on the threshold value. The threshold value is the 95th quantile of the calculated belonging coefficient, 0.06, in order to form a group of nodes with strong relationship.

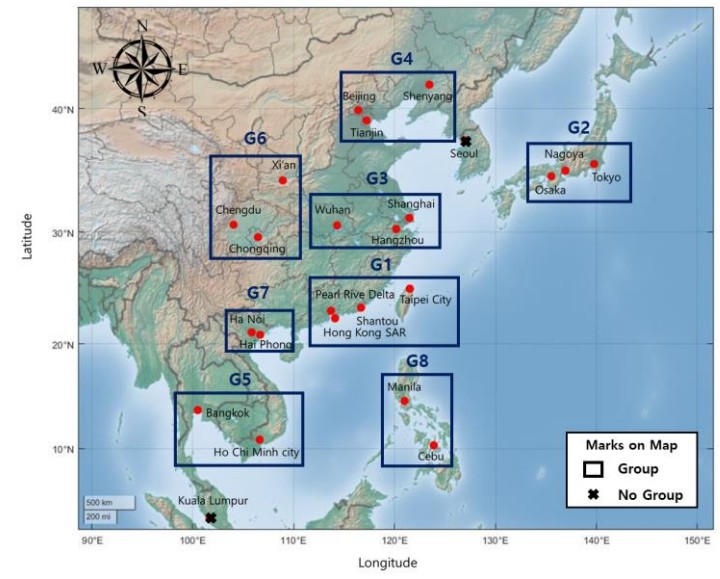

**Figure 5.** Group of nodes using multiresolution community detection; there are 8 groups in the East Asia; G1(Pearl River Delta, Hong Kong SAR, Shantou, Taipei City), G2(Osaka, Nagoya, Tokyo), G3(Wuhan, Hangzhou, Shanghai), G4(Tianjin, Shenyang, Beijing), G5(Bangkok, Ho Chi Minh City), G6(Xi'an, Chengdu, Chongqing), G7(Hanoi, Haiphong), G8(Manila, Cebu); Seoul and Kuala Lumpur did not make group with other nodes.

Nodes in close proximity form a group (Fig. 5). The cities of Seoul (South Korea) and Kuala Lumpur (Malaysia) are not clustered with the others. Seoul has low belonging coefficients with the nearby nodes because of its location in the Korean peninsula. The area is influenced by maritime air mass in summer and continental air mass in winter. Therefore, the precipitation of Seoul is affected by both features and has different characteristics. This feature makes Seoul distinguish between G2 and G4. For the Kuala Lumpur node, the belonging coefficients calculated with other nodes are between 0.03 and 0.05. Unlike other cities, Kuala Lumpur gets significantly influenced by the Boreal winter season and Australia Summer monsoon (Sigh and Xiaosheng, 2020). Therefore, Kuala Lumpur has a different rainfall pattern, and it makes lower results in the belonging coefficient. For evaluating the cluster result, we calculated the Newman-Girvan modularity. We set the divided by 2.5% intervals from 95% to 75%. In the modularity result, 95% show the largest modularity (Table 3). In the Table 3, there are thresholds having same value of modularity. Those thresholds have same result of clustering.

**Table 3.** Result of Newman-Girvan modularity according to threshold

| Threshold | 95% | 92.5% | 90% | 87.5% | 85% | 82.5% | 80% | 77.5% | 75% |
|---|---|---|---|---|---|---|---|---|---|
| Modularity | 0.0313 | 0.0311 | 0.0311 | 0.0311 | 0.0309 | 0.0309 | 0.0294 | 0.0286 | 0.0286 |

**4.4 Relationship between node groups**

Nodes are grouped based on their belonging coefficients (Section 4.3). The relationships between the groups are determined using cross-mutual information analysis. The cross-mutual information is a methodology for calculating MI by adding time lags between targets. It can estimate an appropriate correlation coefficient by considering the time intervals for geographically distant points. In this study, the time lag ranges from −10 to 10 days, and we check the maximum cross-mutual information value and corresponding time lag of each group.

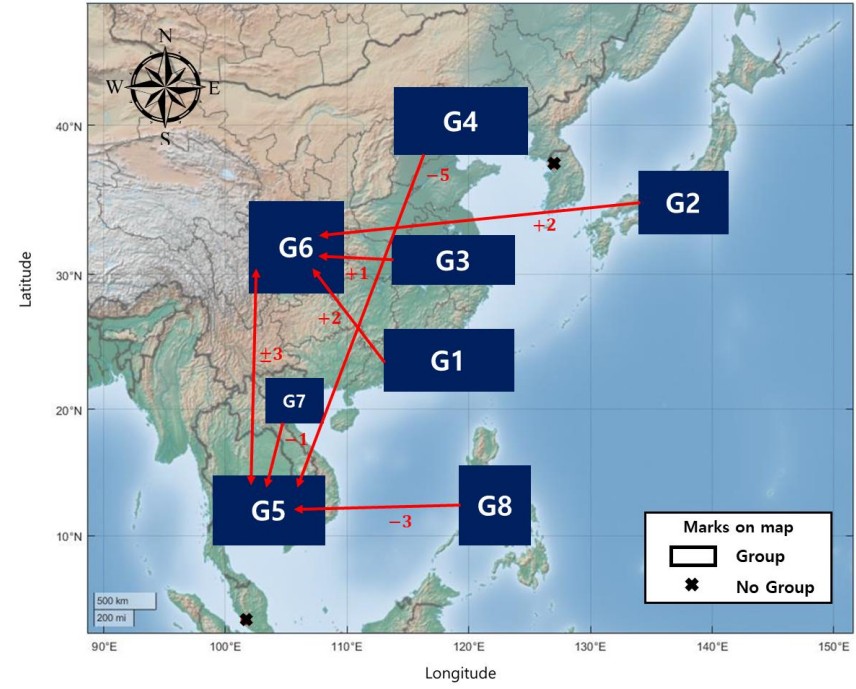

**Figure 6.** The maximum cross mutual information relationship and its time lag value; each arrow points out the maximum relationship group and the numbers under the arrows express the lag time(days) of the maximum cross mutual information value; the figure shows relationship of groups and influence time intervals in East Asia;

   As Figure 6 shows, most groups have strong relationships with G5 or G6 with maximum cross-mutual information values. G5 and G6 have the maximum cross-mutual information value with each other, and this value is larger than other cross-mutual information results. This result indicates that the two regions have a comparatively high relationship. A comparison of the lag times that forms the maximum cross-mutual information indicated that the maximum values are in an interval of fewer than five days. Therefore, East Asian regions can meaningfully relate to each other in a window of five days.

   For founding reasons of rainfall relationship in East Asia, it need to analyze East Asian summer rainfall system. Because East Asian summer rainfall contain more 90% of total rainfall in East Asia (Chen et al., 2015). The relationships in figure 6 are derived from synoptic atmospheric circulation in East Asia. Indian and East Asian monsoons are major factors affecting rainfall in East Asia. The Indian monsoon brings the highly humid wind from the sea that conveys a large quantity of vapor across India and the Bay of Bengal to East Asia. This can reach the northern part of China (Wu, 2017). If the Indian monsoon is strong, a large amount of rainfall can occur in India and northern China. This characteristic is observed in the relationship between G5 and G4 (G5 is the first place affected by the Indian monsoon in East Asia, and G4 is the one in northern China). Water vapor from Indian monsoon moves northwest from the Bay of Bengal, passing mainland China into the Sea of Okhotsk, which is located between the Russia's Kamchatka peninsula and Japan's island of Sakhalin. G5, G6, and G7 in this pathway are related to each other by the Indian monsoon. The movement of water vapor from Indian monsoon is caused by low level jet stream from the Somalian coast. The effect of the South China Sea, which supplies vapor to the mainland, is due to the relationship between G1 and G6. In the summer, the South China Sea vapor causes much rainfall in southern China and arrives in the mainland (Kanaly et al., 1996). Like the Indian Monsoon, the East Asian monsoon affects East Asian rainfall. The East Asian monsoon begins in the Western Pacific, moves eastward through Indonesia, and ends in Japan and South Korea. If it is strong, it affects southern Vietnam and Thailand (Rehe and Akimasa, 2002). This is observed in the relationship between G8 and G5. In the summer, there is an anomalous anticyclone between China and Korea. The anomalous anticyclone locates in the western sea forms a clockwise wind cycle throughout China, Korea, and Japan (Rengquang, 2017). This creates a wind

cycle that transports vapor from Japan to the east and the center of China. This phenomenon forms relationships between G2
and G6 and between G3 and G6.

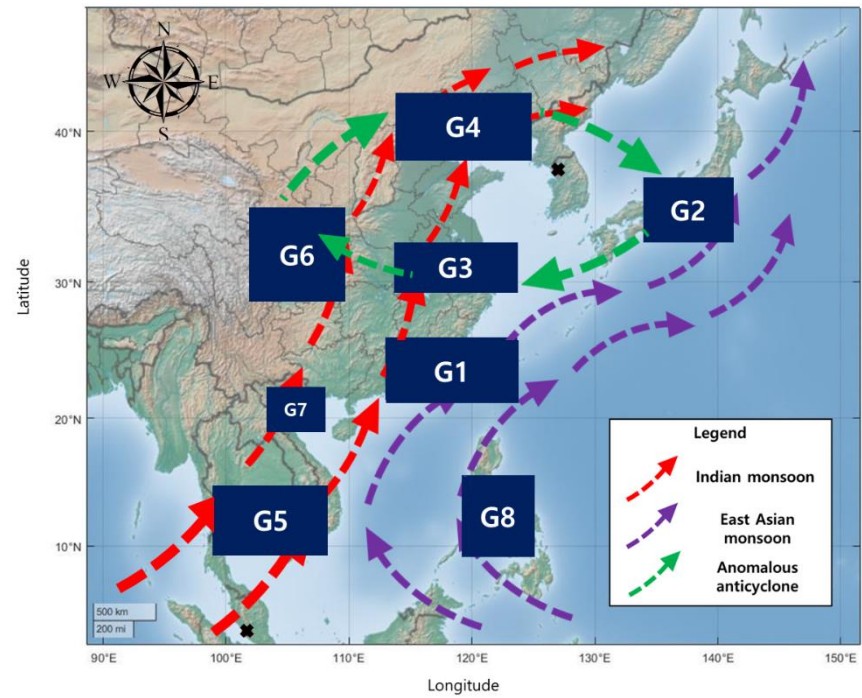

**Figure 7.** Major water vapor transport routes in East Asia; the routes could explain the reasons why the relationship of groups was made li
ke Figure 6; Indian monsoon brings vapor from Indian Ocean, East Asian monsoon gets vapor from Pacific Ocean and East China Sea; An
omalous anticyclone provide vapor in East China, Korea and Japan;

## 4.5 Discussion

Complex network analysis has the advantages of reducing complex phenomena or systems to a graph form, making it easier
to determine characteristics. In addition, it can be used to analyze the effects of network components, perform clustering
analysis. Given these merits, we used complex network analysis to examine the relationships between major cities in East Asia.

To create a rainfall network, we first calculate the MI between nodes and use it as the link's weight. Thus, the network can
reflect the relationship of rainfall in each city and is used as the most important factor in subsequent analyses. In the mutual
information result, we could find that many cities had the lowest value with Tokyo and Beijing. We tried to find why this result
came out, but we could not find any differences in the rainfall data. Therefore, future studies should collect and analyze other
climate and geographical factors to discover unique rainfall characteristics in Tokyo and Beijing.

The adjacency information entropy was calculated and compared to check the effects of nodes in the network. The results
indicate that nodes surrounding the South China Sea and node at the beginning of two monsoons (Indian and East Asian) were
highly ranked, and node's location is one of the essential factors in identifying vital nodes. In the rainfall complex network
research, the high rank nodes mean that they are important sites for the propagation of rainfall event. Based on the interpretation
of high-rank node, we verified that the South China Sea and two monsoons are the primary moisture sources in East Asia. The
South China Sea supplies a huge amount of moisture into East Asia, and the two monsoons pass through it. Vapour from the
South China Sea first affects coastal cities and then moves to other cities in the continent. Thus, rainfall from some cities
affects the neighboring cities. Based on this phenomenon, cities in the South China Sea ranked high. Kuala Lumpur and Manila
are also in high ranks. They have a common thing their location is at the beginning of each major monsoon influence. The two
monsoons pass Kuala Lumpur and Manila first and then move like Figure 7. Because of these characteristics, high-rank nodes
have higher rainfall intensity and the number of rainfall days compared to other cities. On the other hand, low-rank cities have

low rainfall intensity and the rainfall days. Because the low-rank cities commonly are located in the north which is very far from the moisture sources, the low rank cities get less moisture and have less similarity to other cities. Through vital node identification, we could find the major rainfall influence elements in East Asia and it help to interpret the relationship between groups in Section 4.4.

As described in Section 4.3, the belonging coefficient of each node was calculated by using the link weight. Each group consists of nodes located nearby, and their coefficients are significantly higher than those of the other nodes. We also tried to find common physical factors between nodes in the same group. We found that nodes of some groups are located in the same basin or share the river. However, these factors do not apply to all groups. The cluster analysis result is conducted based on the similarity of rainfall. Rainfall is a meteorological phenomenon caused by a combination of various factors such as geographic, hydrologic, meteorological and ecological elements. If two regions have high similarity in rainfall, they share similar characteristics of factors influencing rainfall. The study tried to find common factors that make groups, but failed to find them. These factors are essential to predict concurrent floods in multiple locations. Therefore, in the future study, we will try to find what factors make the high similarity between nodes in the same group by using geographic, meteorological, and hydrological data. After clustering, we apply cross-mutual information analysis to determine the relationships between groups. During the analysis, the lag time is considered because the groups are geographically separated. The cross-mutual information results are interpreted using the rainfall characteristics of East Asia. Two monsoons (Indian and East Asian monsoons) and anomalous anticyclones affect group relationships. This result could also be confirmed when creating groups using various thresholds in Section 4.3. When threshold became 92.5%, Group 5 and Group 7 were merged. After 85%, Group 3 and Group6 became same group. In 80% and 77.5%, Group 8 was added to Group 5+7 and then Kuala Lumpur add to Group 5+7+8. As you can see in Figure 6. Group 7 and Group 8 has strong relationship with Group 5 because of the two monsoon and Group 3 and Group 6 also have high relationship because of anomalous anticyclones. An intriguing result is the strong relationship between G5 and G6. Even with G7 between them, they have a strong relationship. Previous research has primarily focused on the relationship between southern China (G1) and regions surrounding the East China Sea (G5, G7, and G8) (Yuan and Qie, 2008, Hu et al., 2014, Zhao et al., 2017). These studies analyzed the effects of monsoons in the East China Sea but did not expand the region to G6. Therefore, research into the physical interpretation of the link between the G5 and G6 regions is required.

The complex network facilitates a simple analysis of the relationship between East Asian cities. Unlike previous studies, we incorporate temporal factors in the relationship. Through this, we discover new relationships and characteristics of rainfall in East Asia. During the analysis, vital node identification helped to identify important sites for propagation of rainfall and major moisture sources in East Asia. The vital node identification is a useful method to analysis the system or phenomena. Therefore, it can be used for researching natural disaster or meteorological system. Multiresolution community detection method found cities having similar rainfall characteristics according to threshold. It made the groups having similar characteristics and simplified the rainfall network. This helped to understand rainfall relationship between the East Asia major cities. Two methods draw out different results but, they help to interpret the results. For example, the major moisture sources found by vital identification helped to explain the relationship between groups. In addition, their results ultimately helped to understand the rainfall system in East Asia. The results verify that our research framework using a complex network is a helpful process for studying relationships and weather system in regions. The frame contains not only topological analysis, but also statistical analysis and considers temporal factors. Also, in the end, it reflects climate cycle factors and reveals their characteristics.

## 5 Conclusions

Concurrent floods in East Asia inundate the firm's production facilities at multiple locations simultaneously, causing supply chain disruptions at the global level. In this study, we analyzed the spatial relationships of rainfall between major cities in East

Asia using a complex network. The East Asia rainfall network comprises major cities (nodes) and mutual information (links). Once created the network, vital node identification and multiresolution community detection were conducted using adjacency information entropy and multi-community detection. Cross-mutual information defined relationships between cluster groups in East Asia. The results revealed that the network reflected the rainfall characteristics of East Asia and the relationships

significantly affected vital nodes and clustering analysis. In addition, we observed that Southeast Asia and northwest China have a strong relationship. The study observed that although the computational burdens of implementing complex network analysis is not so high, the method accurately reflects the relationship between regional rainfall and can be used to analyze the relationships between various weather factors. In a subsequent study, we intend to evaluate the applicability of complex network methodology to interpret key climate factors, such as ENSO, IOD, and NAO, which have complex interconnection

characteristics.

**Code/Data availability**

The Asian Precipitation – Highly Resolved Observational Data Integration Toward Evaluation (APHRODITE) grid precipitation dataset is available online at APHRODITE's Water Resources (http://aphrodite.st.hirosaki-u.ac.jp/).

**Author contributions**

K. Kim, J. Jung and S. Kim designed and conducted the research. K. Kim analyzed the results, with feedback from H. S. Kim, M. Haraguchi, and S. Kim. K. Kim prepared the paper, with contributions from all co-authors.

**Competing interests**

The authors declare that they have no conflict of interest.

**Acknowledgements**

This research was supported by a grant(2021-MOIS36-002) of Technology Development Program on Disaster Restoration Capacity Building and Strengthening funded by Ministry of Interior and Safety (MOIS, Korea).

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
