# Peer review of "Rainfall patterns in the 24 East Asian mega-cities using a complex network"

_Hydrology and Earth System Sciences, 2021_

## Author Comment (AC1)

**Reviewer #1:**

**Comments and suggestion**

In this study, Kim et al. analysis the spatial dependence of precipitation in 24 cities in East Asia through the complex network framework and explain the results according to the major atmospheric patterns in the area. The manuscript is well structured, and the conclusions are based on sufficient analysis of results. However, while I appreciate the application of mutual information and multiresolution community detection in East Asia, I have concerns about the novelty of using networks to study precipitations, but also on the adherence of the paper to many of the formal rules of scientific writing. Moderate to major revisions are needed for the acceptance according to the following comments.

Thank you very much for your time and insightful review. We have revised the manuscript attentively in order to include your comments. We believe that this manuscript is substantially improved as a result of the revision. Please check our point-by-point response (in blue) to your comments (in red).

<Major comments>

1) As also the authors point out, the complex network framework is widely used to analyse the spatial dependence of precipitation. Thus, its use does not provide novelty. The authors should better clarify the original contribution of the study. Has the study area been analysed by complex network yet? Are mutual information and multiresolution community detection novel frameworks? Has the study ground-breaking results?

➢ Thank you for your good comment. As you mentioned, novelty is fundamental in a research paper. We believe that this study has several novelties. First, while previous complex network studies on precipitation only treated spatial connectivity, we account for both spatial and temporal factors. It made better results in the East Asia relationship, which reflect weather cycle characteristics. Second, we applied new methods (i.e., vital node identification and multiresolution community detection). Those help to analysis subjects in a less complicated and faster manner. Finally, the research framework proposed in this study helps study spatial and temporal connections in large scale regions. From complex network to cross mutual information, the framework contains topological analysis, statistical analysis and also consider temporal factor. This reveals climate connections in regions and reflects weather cycle characteristics.

- (Introduction) We assessed the effects of each region through centrality analysis and grouped the regions according to clustering analysis. Subsequently, mutual information (MI) was calculated with a time lag (i.e., cross-mutual information) to identify the relationships between each group. In previous complex network studies, they have only considered spatial factors for precipitation analysis. It is available to obtain a good result in a small area, but there is a limit to its applicability in a large area. Because there is a significant time difference between two locations in a large

area, therefore, this study proposes an efficient methodology that can evaluate not only spatial but also temporal factors. It also provides a clue to finding the trigger locations of the climate cycle in an area

- (Discussion) The complex network facilitated a simple analysis of the relationship between East Asian cities. Unlike previous studies, we considered temporal factors in the relationship. Through this, we observed new relationships and characteristic of rainfall in East Asia. Two methods (vital node identification and multiresolution community detection) were very useful for analyzing the network and making reliable results. The research results show that our research framework is helpful for studying relationships in regions. The frame contains not only topological analysis but also statistical analysis and considers temporal factors. Also, in the result, the frame reflects climate cycle factors and reveals its characteristics.

2) I do not understand the significance of the representatives' selection from groups. Why do the authors select these? Are the nodes used in the following analyses? Why are they important?

➢ We decided to delete the significance of the representatives' selection from groups. Initially, we used the selection of representatives in the groups for future research in this paper. We plan the future study, which will apply complex network analysis to the whole world. In this case, there will be many nodes and links, making it hard to analysis and interpret a network. Therefore, it will be beneficial to use representatives of the groups, instead of all nodes, because representatives are the most influential nodes in the groups and have characteristics of the groups. We needed to check the validity of the method and applied it in this study. The results show that it has a similar result with vital node identification. As a result, we thought that the selection of representatives would be helpful in a future study. However, we did not use the representatives after selecting them because the discussion is minor.

3) Line 178: I am not sure about the reason why Seoul does not belong to any cluster. The authors explain that the distance is great from other nodes, but the distance between Seoul and Shenyang is 560 km and it is less than the distance from this latter city and Beijing (about 630) and they belong to the same cluster. Can the authors better justify the result?

➢ Your comment was constructive in insight into our research again. After we got your comments, we analysed why Seoul did not make a group with other nodes. We applied event synchronisation, which helps to compare the occurrence pattern of precipitation. Event synchronisation results with Seoul and near cities (Beijing, Tianjin, Shenyang, Osaka, Nagoya, Tokyo) had low value than average event synchronisation reslults of all cities. Through this, it was confirmed numerically that Seoul has a different precipitation pattern from the surrounding cities. The reason for the event synchronisation result is that Seoul is located on the peninsula. The Korean peninsula is

influenced by maritime air mass in summer and by continental air mass in winter. Therefore, characteristics of precipitation in the Korean peninsula are affected by continents and oceans' features and has differences from those. This makes Seoul had a low belonging coefficient with the G2 group (affected by the ocean) and G4 groups (affected by continents). We deled the distance reason and added this abovementioned new reason to the paper.

- Because of the location of Seoul, it had low belonging coefficients with nearby nodes. Seoul is in the Korean peninsula. It is influenced by maritime air mass in summer and by continental air mass in winter. Therefore, the precipitation of Seoul is affected by both features and has different characteristics. This feature made Seoul distinguish between G2 and G4.

4) The references should follow the journal's guidelines: the first name initials after the last name. Please, check the references: the authors have often exchanged last and first names.

➢ We rechecked the guidelines of HESS. Then, we checked all references and revised errors in references.

5) Tables and maps are redundant. It would be better if the authors summarise the information in a single figure, rather than duplicate the results in a table and a map. For example, table 3 and figure 4 can be summarised in a map in which a continuous colour scale could represent the adjacency information entropy values and different sizes of points could represent the rank. Even table 4 - figure 5 and table 6 - figure 6 are redundant.

➢ We revised the table and figures accordingly as below.

- About Table 3 and Figure 4, we made them as one figure like below.

[Figure]

**\<After\>**

[Figure]

**Figure 4. Adjacency information entropy value of cities: Color and size of circle are respectively proportion to the entropy and rank; Ride side of bar shows the adjacency information entropy values of nodes. Except for Taipei city, nodes near south china sea had higher values.**

- About the Table 4 and Figure 5, we deleted Table 4 and put only Figure 5. Readers can find nodes in groups from the figure.

**\**

Table 1. Groups of nodes and their components

| Group | Components |
|---|---|
| G1 | Pearl River Delta, Hong Kong SAR, Shantou, Taipei City |
| G2 | Osaka, Tokyo, Nagoya |
| G3 | Wuhan, Shanghai, Hangzhou |
| G4 | Shenyang, Beijing, Tianjin |
| G5 | Bangkok, Ho Chi Minh City |
| G6 | Chengdu, Xi'an, Chongqing |
| G7 | Ha Noi, Hai Phong |
| G8 | Manila, Cebu |
| Other | Seoul, Kuala Lumpur |

[Figure]

**\<After\>**

[Figure]

**Figure 5. Group of nodes using multiresolution community detection; There are 8 groups in the East Asia; G1(Pearl River Delta, Hong Kong SAR, Shantou, Taipei City), G2(Osaka, Nagoya, Tokyo), G3(Wuhan, Hangzhou, Shanghai), G4(Tianjin, Shenyang, Beijing), G5(Bangkok, Ho Chi Minh City), G6(Xi'an, Chengdu, Chongqing), G7(Hanoi, Haiphong), G8(Manila, Cebu); Seoul and Kuala Lumpur did not make group with other nodes.**

- About Table 5 and Figure 6, we erase Table 5 and only put Figure 6. In this part, the most important thing is that what group has a strong relationship with each group. Figure 6 is the suitable than Table 5 for showing this.

**\**

Table 1. Maximum cross-mutual information and lag time

|  | G1 | G2 | G3 | G4 | G5 | G6 | G7 | G8 |
|---|---|---|---|---|---|---|---|---|
| G1 | - | 0.312 (−3) | 0.333 (2) | 0.312 (2) | 0.438 (−9) | 0.472 (2) | 0.430 (−1) | 0.388 (3) |
| G2 | 0.312 (3) | - | 0.343 (1) | 0.273 (2) | 0.374 (−8) | 0.398 (2) | 0.323 (−9) | 0.333 (−10) |
| G3 | 0.333 (−2) | 0.343 (−1) | - | 0.271 (0) | 0.369 (−9) | 0.496 (1) | 0.355 (−3) | 0.332 (−6) |
| G4 | 0.312 (−2) | 0.273 (−2) | 0.271 (0) | - | 0.409 (−5) | 0.403 (1) | 0.361 (−1) | 0.344 (−10) |
| G5 | 0.438 (9) | 0.374 (8) | 0.369 (9) | 0.409 (5) | - | 0.607 (3) | 0.551 (1) | 0.564 (3) |
| G6 | 0.472 (−2) | 0.398 (−2) | 0.496 (−1) | 0.403 (−1) | 0.607 (−3) | - | 0.535 (−2) | 0.486 (4) |
| G7 | 0.430 (1) | 0.323 (9) | 0.355 (3) | 0.361 (1) | 0.551 (−1) | 0.535 (−2) | - | 0.432 (−5) |
| G8 | 0.338 (−3) | 0.333 (10) | 0.332 (6) | 0.344 (10) | 0.564 (−3) | 0.486 (6) | 0.432 (5) | - |

[Figure]

**\<After\>**

[Figure]

Figure 6. The maximum cross mutual information relationship and its time lag value based on Table 6; Each arrow points out the maximum relationship group and the numbers under the arrows express the lag time(days) of the maximum cross mutual information value; The figure shows relationship of groups and influence time intervals in East Asia;

<Minor comments>

About grammar and sentences, we checked the whole paper by own self and got help from a native speaker.

1) Lines 63-64: "This is because the weights used as input data in each analysis enabled the relationship between regions to reflect in the network and be analysed.";
   ➢ The sentence was re-written as below

   - Because the weights were used as input data in each analysis, network analysis could reflect relationship between regions.

2) Lines 103-104: "Generally, actual systems such as transportation systems or the Internet do not require links to be defined. However, if uncertainty occurs in the connection, researchers must define them.";
   ➢ The sentence was re-written as below

   - Generally, it is easy to define links in systems such as transportation or power grid systems, which have clear physical connections between elements. However, if uncertainty occurs in the connections like social networks, researchers must define them.

3) Lines 154-155: "Various cities had maximum weights for each node, whereas the minimum weights were restricted to a few cities.".

➢ The sentence was re-written as below

- Each node had a maximum value with several different cities, while the minimum value was for certain cities such as Beijing and Tokyo.

4) Line 43: the first mathematician who formulated complex network theory was Leonhard Euler, in 1735.
   ➢ The sentence was re-written as below

- Complex network theory, developed by Leonard Eüler in 1735, expresses and analyses a subject or phenomenon as a graph.

5) Figure 1 and Figure 5: delate "urban" from the labels.
   ➢ We deleted urban in Figure 1 and Figure 5 (Figure 5 is in major comments 5).

[Figure]

Figure 1. Selected 24 major cities in East Asia

6) Section 3.3: the symbols in the formula are not well defined. The authors should better clarify the meaning of the symbols used. For example, what do $v_i$, $v_j$, $c_u$, and nj mean?
   ➢ We added explain about symbols like $v_i$, $v_j$ and $c_u$.

- First, calculate the link intensity ($I_P$) of each link.

$$I_P(e_{ij}) = \begin{cases} \sum_{p=1}^{P} \alpha_p \times \dfrac{\sigma(path_p(v_i, v_j))}{\min(w_i, w_j)}, & e_{ij} \in E \\ 0, & otherwise \end{cases} \qquad (5)$$

Here, $\sigma(path_p(v_i, v_j))$ is the sum of link weights in the path through p links from node i ($v_i$) to node j ($v_j$), P is the parameter of the path, and $\alpha_p$ is a

polygonal effect parameter. For edge $e_{ij}$ between node i and node j, $w_i$ and $w_j$ are their respective strengths.

Second, identify the links with link intensity greater than the selected threshold and create a group of nodes with the identified links.

$$v_j \in V, \qquad I_P(e_{ij}) > t$$
$$v_j \in c_u, \qquad I_P(e_{ij}) > t \tag{6}$$

Here, t $(0 < t \le 1)$ is the selected threshold, and $c_u$ is a group of nodes.

Third, calculate the belonging coefficient $(I_P)$ of the nodes in node set u $(c_u)$.

$$I_P(c_u, v_j) = \sum_{v_i \in c_u} I_P(e_{ij}) \tag{7}$$

---

## Author Comment (AC2)

**Reviewer #1:**

**Comments and suggestion**

This is a short and sweet, very well laid out article with good explanations, development and description. It introduces a new idea (to me) called Complex network analysis, which is well introduced and developed. I enjoyed the sequence of figures concluding in an evocative image in Figure 7. The tables and Figures were well presented and organised.

From my point of view it is a go, once the suggested corrections are all annotated on the article, which I am returning with this review. I am highlighting the less trivial suggestions for correction below my signature, which is my wont. The annotated article is being returned for correction with my review.

Thank you very much for your time and insightful review. We have revised attentively the manuscript in order to include your comments. We believe that this manuscript is substantially improved as a result of the revision. Please check our point-by-point response (in blue) to your comments (in red).

<Major comments>

1) please expand your figure caption to tell us what the figure is trying to show the reader, before searching in the text. This comment applies to all figures and some of the more complex tables

➢ Thank you for your good comment. We totally agree with your comment. Therefore, we added more detail explanation about Figure and Table in Captions. We hope this will help readers understand Figure and Table.
- Table 1. Basic statistics values for rainfall data of cities
  ⇒ Table 1. Basic statistics values for rainfall data of cities; Basic statistics contain average, standard deviation, coefficient of variation and skewness;
- Table 2. Average, maximum, and minimum link weights of each node
  ⇒ Table 2. Average, maximum, and minimum link weights of each node; Parentheses under link weights is nodes that forms a maximum or minimum value for target node;
- Figure 4. Ranks of nodes using vital node identification
  ⇒ Figure 4. Adjacency information entropy value of cities; Color and size of circle are respectively proportion to the entropy and rank; Ride side of bar shows the adjacency information entropy values of nodes; Except for Taipei city, nodes near south china sea had higher values;
- Figure 5. Group of nodes using multiresolution community detection
  ⇒ Figure 5. Group of nodes using multiresolution community detection; There are 8 groups in the East Asia; G1(Pearl River Delta, Hong Kong SAR, Shantou, Taipei City), G2(Osaka, Nagoya, Tokyo), G3(Wuhan, Hangzhou, Shanghai), G4(Tianjin,

Shenyang, Beijing), G5(Bangkok, Ho Chi Minh City), G6(Xi'an, Chengdu, Chongqing), G7(Hanoi, Haiphong), G8(Manila, Cebu); Seoul and Kuala Lumpur did not make group with other nodes.

- Figure 7. Major water vapor transport routes in East Asia; The routes could explain the reasons why the relationship of groups was made like Figure 6

⇒ Figure 7. Major water vapor transport routes in East Asia; The routes could explain the reasons why the relationship of groups was made like Figure 6; Indian monsoon brings vapor from Indian Ocean, East Asian monsoon gets vapor from Pacific Ocean and east china sea; Anomalous anticyclone provide vapor in east china, Korea and Japan;

2) this last figure '7' is in microns = $1m^{-6}$. This is spurious - try 3 significant figures?

➢ In Figure 7., Max mi means Maximum cross-mutual information value. We want to express relationship between groups based on the maximum cross-mutual information result. However, Max mi in Legend could make confsuion on readers. Therefore, we fixed Legend in the Figure 7. And wrote more explaination in Caption.

[Figure]

Figure 6. The maximum cross mutual information relationship and its time lag value; Each arrow points out the maximum relationship group and the numbers under the arrows express the lag time(days) of the maximum cross mutual information value; The figure shows relationship of groups and influence time intervals in East Asia;

3) is this $k_j$ an inverse-distance weight, like kriging?

➢ $k_j$ means 'Degree' which is basic index in complex network analysis. The degree is the number of nodes which have links with target node in unweighted network. About the weighted network, it is sum of weight of links that have connection with target node. Reader need to know about degree for understand vital node identification method. Therefore, we put explanation about degree in equation (2).

- First. Calculate degree($k_j$) of each node in the network

$$k_i = \sum_{j \in \Gamma_j} w_{ji} \tag{2}$$

Here, $\Gamma_j$ is a group of nodes that form links with node j. $w_{ji}$ is weight of link that connect node j and node i. If a network is unweighted, degree is the number of neighbor nodes.

Second, calculate the adjacency degree ($A_i$) of each node.

$$A_i = \sum_{j \in \Gamma_j} k_j \tag{3}$$

Third, calculate the selection probability ($P_{i_j}$).

$$P_{i_j} = k_i / A_j \tag{4}$$

Final, calculate the adjacency information entropy ($E_i$).

$$E_i = \sum_{j \in \Gamma_j} (P_{i_j} log_2 P_{i_j}) \tag{5}$$

After comparing the calculated adjacency information entropy of each node, the importance is determined according to the descending power.

4) Comments on References

➢ We checked guidelines of HESS again. Then, we checked all references and revised errors in all references.

- East Asia accounts for 54% of the global supply chain, providing a wide range of services and products across the world (Ann et al, 2020).
- Wang, Z., Mu, J., Yang, M., and Yu, X.: Reexaming the mechanisms of East Asian summer monsoon changes in response to non-East-Asian anthropogenic aerosol forcing, Journal of Climate, 33(8), 2929-2944, https://doi.org/10.1175/JCLI-D-19-0550.1, 2020.

---

## Referee Report (RR1)

I read with interest the answers to the review. The responses are appropriate and improve the manuscript. However, there are still some issues about the drafting that have to be addressed, thus minor revisions are necessary.

**Minor comment:**

In order to have a more legible text, it would be better if the results and discussion were reported in the present tense. In the next part, I report some phrases that should be checked.

Line 62: "Because the weights are used as input data in each analysis enabled the relationship between regions to reflect in the network and be analyzed." The phrase is not clear.

Line 65: "Before related complex network studies, they only considered spatial factors. But in this study, we added temporal factors in relationship." The phrase can be rewritten as follows: "Past studies about complex network considered only spatial factors, while here we add also temporal factors.

Line 71: "In this study, the major cities in East Asia were studied." The phrase can be rewritten as follows: "In this analysis, the major cities in East Asia are studied." Or: "In this study, the major cities in East Asia are analyzed."

Line 105: "The most widely used methodology is the correlation coefficient." Reference.

Line 107: "While various previous studies used the Pearson correlation coefficient for links, they tend to derive inaccurate values if they were applied to nonlinear data." Reference.

Line 119: "While some methods of identifying important nodes have been developed, these methods have limitations and are only 120 applicable to certain types of networks." Reference.

Lines 123-125: If you are computing the degree accounting for the weight of links, the actual name is strength. Degree defines just the number of neighbors, as you specified.

Line 151: "In this study, we described the shape of a rainfall network as being weighted and undirected." The phrase should be rewritten as follows: "In this study, we design a rainfall network as a weighted and undirected graph. "

Line 178: "To form a group of nodes with strong relationships, the threshold value was the 95th quartile of the calculated belonging coefficient, 0.06." The phrase should be rewritten as follows: "The threshold value is the 95th quartile of the calculated belonging coefficient, 0.06, in order to form a group of nodes with strong relationships."

Line 186: ". Because of location of Seoul, it had low belonging coefficients with nearby nodes. Seoul is in the Korean peninsula." The phrase should be rewritten as follows: "Seoul has low belonging coefficients with the nearby nodes because of its location in the Korean peninsula. The area …."

Line 189: Why has Kuala Lampur belonging coefficient so low?

Line 205: "The relationships in figure 6 were derived from the characteristics of East Asian rainfall." I am not sure about the meaning of this phrase: do you mean that the characteristics of the rainfall cause the relationship between the groups? Or do you mean that the relationship between the groups can be explained by the synoptic atmospheric circulation, as you explain later? Please, clarify.

Line 230: "To check the effects of nodes in the network, the adjacency information entropy was calculated and compared." The phrase can be rewritten as follows: "The adjacency information entropy was calculated and compared to check the effects of nodes in the network."

Line 259; ". After the network was created" the phrase should be rewritten as follows: "Once created the network,"

---

## Referee Report (RR2)

[revised manuscript text omitted]

**3.2 Vital node identification using adjacency information entropy**

Important nodes in a network have various effects on the structure or function of the network. Identification of these nodes is of practical and theoretical value (Xiang et al., 2020). For example, if a government identifies which places have a key role in power grids and traffic networks, it can effectively invest and create defense measures to prepare for blackouts and traffic jams. While some methods of identifying important nodes have been developed, these methods have limitations and are only applicable to certain types of networks. Xiang et al. (2020) developed a new methodology based on information entropy, making it applicable to all types of networks. This method has better results than the previously existing methods. The procedure of the method is as follows:

First. Calculate degree($k_j$) of each node in the network

$$k_i = \sum_{j\in\Gamma_i} w_{ji} \qquad (2)$$

Here, $\Gamma_j$ is a group of nodes that form links with node j. $w_{ji}$ is weight of link that connect node j and node i. If a network is unweighted, degree is the number of neighbor nodes.

Second, calculate the adjacency degree ($A_i$) of each node.

$$A_i = \sum_{j\in\Gamma_j} k_j \qquad \boxed{\textbf{AGAIN}, \text{ is this an inverse-distance weight, like Kriging? Please elaborate}} \qquad (3)$$

[revised manuscript text omitted]

---

## Referee Report (RR3)

In the manuscript, the authors use an approach based on complex networks to analyse rainfall patterns in 24 mega-cities of East Asia. The choice of the cities is based on their importance based on the degree of urbanization and their vulnerability to extreme rainfall. They used mutual information and clustering to assess the importance of the cities in terms of their strategic location and also a temporal lag based analysis to recognize how the impacts of rainfall on the cities are inter-related. The approach is novel and successfully brings to light many important relationships between the rainfall characteristics of the different cities, some of which needs to be climatologically interpreted in future research. However, I have few major revisions based on the questions related to the interpretation of results, which I feel needs to be addressed, and some more minor revisions listed below. I recommend the publication of this manuscript in HESS after the revisions are made.

**Major revisions:**

1. The authors rank the importance of the nodes (cities) based on the values of adjacency information entropy and average link weight. They have explained why some locations have a low rank. However, there is no physical interpretation of the cities having very high/low entropy values. Are these cities more prone to flooding due to extreme rainfall from monsoon/typhoons from the South China Sea? Also do they know any reason why the group of cities in the north of the map have low values compared to those in the south? Such an explanation would explain the purpose of conducting the vital node identification methodology.

2. I like how the clustering analysis identify the cities having similar rainfall patterns. However, the authors only explain about Seoul and Kuala Lumpur as to why they are isolated from all groups. There is no interpretation of the cities which form groups. I have a feeling that some groups may be formed because the particular cities lie in the basin of the same river, for instance, G4 is in the Yellow river basin and G3 in the middle/lower reaches of the Yangtze, which may be the cause of similar rainfall characteristics. Such knowledge is important to predict concurrent floods in multiple locations. Could you please check if such is the case of most clusters?

3. I wonder if lowering the threshold value in the clustering analysis, leads to bigger clusters, and unravels grouping together of those clusters which are found to be related in the time lag analysis. For instance, G2, G3, G4 and G6 may be related as they are linked by the anticyclone and two or more of the groups fall in the same cluster when a lower threshold is chosen. Has the authors verified that?

4. Since the authors used the rainfall time series of the whole year for the analysis, how do they know while climatologically interpreting in Sec. 4.4, that the relation between groups is due to Indian monsoon which takes place in summer, or that the anticyclone is not specific to a certain season?

**Minor revisions**

1. The main conclusions of the paper is not clear when reading the abstract. The concluding statement should talk more about the authors' findings.

2. There is a consistent confusion created by the consistent usage of the term 'correlation coefficient' synonymously in place of 'similarity measure'. Since correlation coefficient is a type of linear similarity measure, whereas mutual information is nonlinear, they are different. It is also not clear in the abstract, introduction and methodology Sec. 3.1 therefore whether the authors use mutual information or correlation in their work. Kindly correct it.

3. Figure 5 has a typo in labelling of boxes G2 and G3.

4. In the data, it is not clear if you use a single grid point from APRODITE to represent each city of a group of grid points.

5. Lines 58-59 in the introduction is incorrect. A complex network may be of several types. What the authors seem to refer to are functional climate networks in which connectivity is defined on the basis of 'statistical interdependence' between the pairs of nodes. Please change the sentence.

6. Line 106 in Sec 3.1, a node need not be a fixed element. A node in a complex network is a dynamical unit, it may be fixed, movable, disappearing. In case of climate network, the nodes are spatial grid points which represent a dynamical subsystem. Kindly correct the sentence.

7. Sec. 3.2 in methodology, please add a sentence on the relevance of vital nodes or high degree nodes in climate system.

8. Sentence 'Each node has different link weights' on line 169 is confusing, as link weight is between pairs of nodes. Please clarify or remove.

9. In line 175, authors talk about narrower or wider range of average, maximum and minimum link weights. They can use standard deviation of the range of av., max., min. link weights instead to quantify the 'range'.

10. Section 4.2, please specify that high ranking nodes means nodes of high adjacency information entropy. The term does not appear even once in the explanation.

11. I do not understand in lines 186-187, 'High ranking nodes in center of the maps… low-ranking nodes are diametrically opposed..' Kuala Lumpur, a high ranked node is not in center. 'diametrically opposite' is not clear. Please remove the explanation if it is not needed.

12. Lines 184-185, 'low mean of MI' and 'large average of MI', do you refer to the second column of average link weight in Table 2 ? Please clarify.

13. Line 233 onwards, Indian monsoon does not move northwest, rather the moisture bearing cross equatorial south-westerly low level jet coming from the Somalian coast, which causes Indian monsoon, moves northwest.

14. Replace 'vapour provider' and 'vapour' in line 258 with 'moisture source' and 'water vapour' respectively.

15. Mention 'belonging coefficient' specifically in line 261.

---

## Referee Report (RR4)

This manuscript mainly discusses the application of complex-network-based methods for the spatial and temporal characteristics of rainfall in East Asian region, with vital nodes identification and community detection being used. As far as I know, this is a good direction from the perspective of application. I am not a climate expert, therefore mostly I would like to consider how the methods deliver a solid and convincing application. There are still several main points I would like to emphasize here, hoping that authors in this manuscript would provide proper explanations, both in the later revised manuscript and response report.

**Regarding dataset:**
(1) Why not directly use the original precipitation points (or maybe a slight coarsen resolution, if the original resolution is too small), because I see from Fig. 2 that in East Asian region, there are a lot of points more than 24?
24-city is a quite small-scale. That means the constructed network has only 24 nodes. Even for the selected 24 cities, there are probably several stations inside each of them. Then it would be better to consider the original dataset, because the aggregation of rainfall data for each city is essentially omitting some local information. Besides, as far as I know, the computational complexity (that is, the actual running time) for a network with for example 1000 nodes, can still be pretty small. Another reason to consider a higher resolution is that it delivers a more convincing conclusion.

**Regarding methods:**
(1) Why use vital node identification and community detection together? And how they both contribute to the final conclusion?
In the domain of complex network analysis, vital node identification and community detection are different research problems, although there might be some overlapping between them. I initially thought there might be some connections between the results obtained from these two methods in this manuscript. But it seems to me that they are separate contents. In my opinion, vital nodes are possibly useful for disaster mitigation, while community detection could be useful for finding out regions of similar climatological behavior, but there might be a lack of explanation on why they should be considered together, and specifically, how they help in obtaining the final conclusion.
(2) Regarding the community detection, how is the quality of community structure (or "groups" in this manuscript) measured? Also, how is the threshold "t" determined?
When applying community detection, commonly accepted measures include modularity and normalized mutual information. But there is a lack of explanation in this regard. The threshold "t" could be any value between the range of [0, 1]. Different values could lead to different forms of groups. If there is not a standard way to determine what it would be, then why the group in Fig. 5 would be chosen as the final result.

**Others:**
(1) From the current version of the Abstract, the main conclusion is not emphasized.
(2) Contents between 153 and 165 (both equations and paragraphs) should be reorganized in a more formal way
(3) Fig. 5 has a label problem, two "G2".

---

## Referee Report (RR5)

I read with interest the answers to the review and I think that the manuscript is improved. There is just a couple of phrases that must be checked.

**Minor comment:**

Line 24: "occurred worldwide during 2000 to 2020", it should be: "during the period from 2000 to 2020".

Line 65: "Using the correlation coefficient makes that a network reflect relationship between region": this phrase should be rewritten in a better way.

---

## Referee Report (RR6)

Thanks for the response to all my previous comments. Still there are several minor mistakes in the current version.

1. Please check if is reasonable to use the term 'multi-community detection'. Originally, it should be 'multiresolution community detection'. I don't think this abbreviation delivers the same meaning.

2. Please change Eqs. 6 and 7 in a more formal way. Shouldn't there be parentheses if they are conditional formulas?

3. Please check Eq. 9, make sure it is in the right form. If I remember correctly, the current form is not correct. Should be $g(a) \in P$, isn't it?

---

## Author Response (AR2)

**Author's response**

We got second review from two reviewers during discussion interval. The reviewers talked about many factors in the paper. We tried to reflect comments on revised manuscript. We believe that the paper is substantially improved as the result of the revision. We wrote point-by-point response (in blue) to reviewer's comments (in red). Also, we put revised part of the manuscript in black.

**<Reviewer 1>**

1) In order to a more legible text, it would be better if the results and discussion were reported in the present tense.

   ➢ Following the above comment, we changed the Result and Discussion part as present tense. Because both parts are too long to put in the Author's response, it would be appreciate to check changes of the Result and Discussion part in the revised paper or revised paper with track change.

2) Line 62 is not clear.

   ➢ In Line 62, we tried to express that using a correlation coefficient as a weight makes a network reflect relationship between regions. Because a link weight is one of the most important input value for constructing a network and both analysis (Vital node identification and Multi-resolution community detection) use it as important input data. The current expression about Line 62 is not clear for understanding the meaning. Therefore we changed it like below.

       - (Line 62) Using the correlation coefficient makes that a network reflect relationship between regions. The link weight is one of the most important inputs for constructing and analyzing a network.

3) Some phrase (Line 105, Line 107, Line 119) need refences.

   ➢ We put references on lines 105, 107, and 119 on each line.

       - (Line 105) The most widely used methodology is the correlation coefficient (Donges et al., 2009).
       - (Line 107) While various previous studies used the Pearson correlation coefficient for links, they tend to derive inaccurate values if they are applied to nonlinear data (Zadian et al., 2018).
       ※ Zadian M.A., Haapasilta, V., Relan, R., Paasonen, P., Kerminen, V. M., Junnien, H., Kulumala, M. and Foster, A.S.: Exploring non-linear associations between atmospheric new-particle formation and ambient variables: a mutual information

approach, Atoms. Chem. Phys., 18(7), 12699-12714, https://doi.org/10.5194/acp-18-12699-2018, 2018.

- (Line 119) While some methods of identifying important nodes have been developed, these methods have limitations and are only applicable to certain types of networks (Mester et al., 2021).

※ Mester, A., Pop, A., Mursa, B. E. M., Grebla, H., Diosan, L. and Chira, C.: Network Analysis Based on Important Node Selection and Community Detection, Mathematics, 9(18), 2294, https://doi.org/10.3390/math9182294, 2021.

4) (Line 123-125) If you are computing the degree accounting for the weight of links, the actual name is strength. Degree defines just the number of neighbors, as you secified.

➢ About the comment, we checked exrpession about Degree and Strength in other papers (Newman, 2001, Barrate etal. 2004, Opsahl et al. 2010) and found that our expression needs to be fixed. For a weighted network, it is more suitable to express degree as strength. We appreciate for letting know about the expression.

※ Newman, M. E. J.: Scientific collaboration networks. II. Shortest path, weighted networks, and centrality, physical reveiw E, 64, 016132, doi: 10.1103/PhysRevE. 64.016132., 2001.

※ Barrat, A., Barthelemy, M., Pastor-Satorras, R. And Vespignani, A. : The architecture of complex weighted networks, Proceedings of the National Academy of Sciences, 101(11), 3747-3752, https://doi.org/10.1073/pnas.0400087101, 2004.

※ Opsahl, T., Agneessens, F. And Skvoretz, J. : Node centrality in weighted networks : Generalizing degree and shortest paths, Social Networks, 32(3), 245-551, doi:10.1016/j.socnet.2010.03.006, 2010.

- (Line 135) First Calculate strength($k_i$) of each node in a weighted network

5) (Line 189) Why has Kuala Lampur belonging coefficient so low?

➢ Precipitation of Kuala Lumpur gets alot of influence from the Boreal winter season or Australia Summer monsoon (Sigh and Xiaosheng, 2020). However, the other cities except for Kuala Lumpur, the primary influencer of precipitation is the Indian monsson and East Asia monsoon. Therefore, Kuala Lumpur has different precipitation characteristics and it makes lowering the belonging coefficient of Kuala Lampur. We put this explaination in Line 189 for authors.

- Unlike other cities, Kuala Lumpur is significantly influenced by the Boreal winter season and Australia Summer monsoon (Sigh and Xiaosheng, 2020). Therefore, Kuala Lumpur has a different rainfall pattern, and it makes lower results in the belonging coefficient.

※ Singh, V. and Xiaosheng, Q.: Study of rainfall variabilities in Southeast Asia using long-term gridded rainfall and its substantiation through global climate indices, Journal of Hydrology, 585, 124320, https://doi.org/10.1016/j.jhydrol.2019.

124320, 2020.

6) (Line 205) ″The relationships in figure 6 were derived from the characteristics of East Asian Rainfall″. I am not sure about the meaning of this phrase : do you mean that the characteristics of rainfall cause the relationship between groups ? or do you mean that the relationship between the groups can be explained by the synoptic atmospheric circulation, as you explain later? Please, clarify.

➢ Our intention about Line 205 is more close to later meaning. Most of rainfall in East Asia is concentrated on summer season. And synoptic atmospheric circulation (Indian monsoon and East asia monsoon) is a main influence factor on summer rainfall. Therefore, many previous studies concluded that the synoptic atmospheric circulation makes characteristics of rainfall and rainfall relationship in East Asia. But our expression did not clearly express the intention. So, we changed Line 205 like below.

- The relationships in figure 6 are derived from synoptic atmospheric circulation in East Asia.

7) The phrases (Line 65, 71, 151, 178, 186, 230, 259) need to be rewritten.

➢ We appreciate comments on English expression. We checked not only the lines suggested by the comments but also whole paper.

- (Line 65) Past studies about complex network considered only spatial factors, while we add also temporal factors.
- (Line 71) In this research, the major cities in East Asia are analyzed (Fig. 1).
- (Line 151) In this research, we design a rainfall network as a weighted and undirected graph.
- (Line 178) The threshold value is the 95th quantile of the calculated belonging coefficient, 0.06, in order to form a group of nodes with strong relationship.
- (Line 186) Seoul has low belonging coefficients with the nearby nodes because of its location in the Korean peninsula. The area is influenced by maritime air mass in summer and continental air mass in winter. Therefore, the precipitation of Seoul is affected by both features and has different characteristics.
- (Line 230) The adjacency information entropy was calculated and compared to check the effects of nodes in the network.
- (Line 259) Once created the network, vital node identification and clustering analysis were conducted using adjacency information entropy and multi-community detection.

**<Reveiwer 2>**

1) Where is Centre for Research on the Epidemiology of Disasters (CRED) ?

➢ Centre for Research on the Epidemiology of Disasters has been active for over 40 years in the fields of international disaster and conflict health studies with activities linking relief, rehabilitation and development. The Centre promotes research, training and technical expertise on humanitarian emergencies, particularly public health and epidemiology. CRED's research scope includes natural disasters and crisis caused by civil strife, conflict or others. CRED offers EM-DAT (International Disaster database). This database was developed in 1988 to rationalize decision making for disaster preparedness, while also providing an objective base for vulnerability assessment and priority setting. EM-DAT contains essential core data on the occurrence and effects of disasters worldwide, running from 1900 to the present. If reviewer wants to find more detailed information about CRED, the below link is connected to main web page of CRED

https://www.cred.be/

We added information of CRED in Introduction part for authors who have any knowledge about the CRED.

- (Introduction) According to disaster database of the Centre for Research on the Epidemiology of Disasters (CRED), which offer essential core data on the occurrence and effects of disasters all over the world, annual average of 165 flood disasters occurred worldwide during 2000 to 2020, resulting in 5,278 deaths and economic damage up to $29 million.

2) Check the Refs about Watts and Strogatz (1998) and Barabási and Albert (1999). Also check the rest

➢ We checked the Refs about the two papers and found they were omitted. Therefore, two papers were added in the Reference part. Also we checked other papers whether those were correctly written in Reference part.

- (Reference) Watts, D. J. and Strogatz, S. H.: Collective dynamics of 'small-world' networks, Nature, 393, 440-442, https://doi.org/10.1038/30918, 1998.
  Barabási, A. L. and Albert, R.: Emergence of scaling in random networks, Science, 286, 509-512, https://doi.org/10.1126/science.286.5439.509, 1999.

3) Seperate all paragraphs by space and/or indent

➢ We checked all paragraphs and put space in the beginning of each paragraph

4) This sentence is confusing because there is no mention of rain nor precipitation in this document (This sentence = 'we used selected cities from the World Bank Group report called "East Asia's Changing Urban Landscape" (The World Bank, 2015)')

➢ We agreed with the reveiwer's comment. We changed the sentence like the below accordingly.

- (Study area and materials) Among East Asia cities, we used selected cities selected by Haraguchi et al. (2019). They chose the cities with more than 5 million people and a high degree of urbanization. Because rainfall often causes numerous floods in the region, and with growing urbanization, more and more cities suffer from small and medium-sized frequent floods, as well as large-scale low-probability floods (The World Bank, 2015).
- (Reference) Haraguchi, M., Kim, S. and Lall, U.: Correlated Risks for Heavy Precipitation in Mega-cities in East Asia, in: American Geophysical Union, Fall Meeting 2019, Meeting in San Francisco, U.S.A, 8-14 December, abstract #GC43G-1316, 2019.
  Disaster Risk Management Overview: https://www.worldbank.org/en/region/eap/ brief /disaster-risk-management, last access: 29 May 2015.

5) How did you manage to gather the data from countries outside China ?

➢ In the research, we used the APHRODITE (Asian Precipitation-Highly-Resolved Observation Data Integration Towards Evaluation of Water Resources). The APHRODITE is a long-term daily grid precipitation dataset that combines rain gauge data, remotely sensed data, and geographic information over Asia. International Research Institute for Climate and Society (IRI) at Columbia University provides the data on a 0.25-degree grid for Asia. Therefore, rainfall data of countries outside China were collected from grid data.

6) Please expand your figure cpation tell us what the figure is trying to show the reader, before searching in the text. This comments applies to all figures and some of the more complex tables.

➢ We remembered this comment in the first revision stage. While we changed almost all captions in the table and figures, we did not change the caption in figure 1. In this time, we checked all captions again and revised some captions. We hope we reflect the comment correctly on the captions.

[Figure]

**Figure 1.** APHRODITE data (http://aphrodite.st.hirosaki-u.ac.jp/products.html ) of cities; Dots are example of station data distribution that were used for making data(Red: individual collections, Black: pre-complied data, Blue: global telecommunication networks (GTS) based data obtained from Global summary of day (NCEI/NOAA)). Rectangles show the domain of precipitation data. In this research, precipitation data on a 0.25 degree grid for Monsoon Asia (MA) for the period from 1981 to 2015.

[Figure]

**Figure 2.** Various shapes of networks with the same number of nodes and links; Each network has 4 nodes and 4 links. However, those shows different shapes and have different topological characteristics.

[Figure]

**Figure 3.** Adjacency information entropy value of cities; color and size of circle are respectively proportional to the entropy and rank. The right side of bar shows the adjacency information entropy values of nodes; except for Taipei city, nodes near the South China Sea, which had higher values;

7) Comments about Table 1 : Use 3 significant figure for all

➢ About Table 1, there were several problems. Therefore, we wrote all values in the table with three decimal places and changed the font to 'Times New Roman'.

**Table 1.** Basic statistics values for rainfall data of cities; Basic statistics contain average, standard deviation, coefficient of variation and skewness.

| Station | Average (mm/day) | Standard deviation | Coefficient of Variation | Skewness |
|---|---|---|---|---|
| Pearl River Delta | 4.277 | 9.416 | 2.202 | 4.063 |
| Tokyo | 3.637 | 10.736 | 2.952 | 7.044 |
| Shanghai | 2.985 | 6.446 | 2.160 | 3.993 |
| Beijing | 1.316 | 4.377 | 3.325 | 6.395 |
| Manila | 6.241 | 12.514 | 2.005 | 5.023 |
| Seoul | 3.457 | 9.427 | 2.727 | 5.543 |
| Osaka | 2.802 | 5.900 | 2.105 | 4.619 |
| Bangkok | 1.101 | 4.067 | 3.694 | 7.308 |
| Tianjin | 3.903 | 9.932 | 2.544 | 4.910 |
| Shantou | 2.211 | 4.853 | 2.195 | 5.279 |
| Chengdu | 3.782 | 6.241 | 1.650 | 3.246 |
| Ho Chi Minh City | 4.551 | 11.314 | 2.486 | 4.919 |
| Nagoya | 3.176 | 7.762 | 2.444 | 4.492 |
| Wuhan | 4.583 | 11.278 | 2.461 | 4.427 |
| Hong Kong SAR | 4.889 | 6.460 | 1.321 | 2.177 |
| Shenyang | 1.590 | 4.946 | 3.11 | 6.183 |
| Taipei | 6.955 | 16.128 | 2.319 | 6.192 |
| Hangzhou | 3.614 | 7.413 | 2.051 | 3.981 |
| Kuala Lumpur | 6.196 | 7.970 | 1.286 | 2.311 |
| Xi'an | 1.527 | 4.099 | 2.684 | 5.054 |
| Ha Noi | 4.053 | 8.870 | 2.189 | 4.417 |
| Chongqing | 2.790 | 5.751 | 2.061 | 4.890 |
| Cebu | 4.126 | 6.759 | 1.638 | 4.923 |
| Hai Phong | 3.801 | 9.530 | 2.507 | 5.023 |

8) Why mutual information in equation (1) is expressed as I(A,B), not MI(A,B) ?

➢ The papers (for example, Donges et al. (2009), Ghorbani et al. (2021)) about mutual information usually express mutual information as I(A,B). So we used the same expression in this paper. But if it confuses the readers, we changed I(A,B) into MI(A,B).

$$\text{MI}(A, B) = \sum_{b \in B} \sum_{a \in A} p(a, b) log(\frac{p(a, b)}{p(a)p(b)}) \tag{1}$$

9) Comments about Vital node identification method : is the adjacency degree is same to inverse-distance weight, like Kringing ?

➢ $A_i$ is not an inverse distance weight. It is just total weight of neighbor nodes of node i. Let me explain it with orginal paper (Xu et al., 2020). If there is a sample network like below,

[Figure]

degree of node must be calculated first. This network has weight. Threrefore, the degree is equivalent to the sum of link weight.

| Node | 1 | 2 | 3 | 4 | 5 | 6 | 7 |
|---|---|---|---|---|---|---|---|
| $k_i$ | 0.5 | 0.6 | 0.7 | 0.4 | 0.6 | 0.3 | 1.6 |

By using degree of node, Adjacency degree ($A_i$) can be calculated like below :

$$A_1 = k_2 + k_7 = 0.6 + 1.6 = 2.2$$
$$A_2 = k_1 + k_3 + k_7 = 0.5 + 0.7 + 1.6 = 2.8$$
$$A_3 = k_2 + k_4 + k_7 = 0.6 + 0.4 + 1.6 = 2.6$$
$$A_4 = k_3 + k_5 + k_7 = 0.7 + 0.6 + 1.6 = 2.9$$
$$A_5 = k_4 + k_6 + k_7 = 0.4 + 0.3 + 1.6 = 2.3$$
$$A_6 = k_5 + k_7 = 0.6 + 1.6 = 2.2$$
$$A_7 = k_1 + k_2 + k_3 + k_4 + k_5 + k_6 + k_7 = 0.5 + 0.6 + 0.7 + 0.4 + 0.6 + 0.3 = 3.1$$

As you can see, $A_i$ is sum of degree of neighbors nodes to node i. If node i has many links or links with high weight, it has high value of $A_i$. It would be better put examples of Vital node identification for authors. But this can make paper longer and detailed explanation is already in the reference paper. So we recommend to authors for checking the reference paper like bleow

- The procedure of the method is as follows (more detailed explanation about the Vital node identification method is in Xu et al. (2020)):

First. Calculate degree($k_i$) of each node in the network

$$k_i = \sum_{j \in \Gamma_i} w_{ji} \qquad\qquad - \quad (2)$$

Here, j is the neighbor of node i. $\Gamma_i$ is the set of neighbors of node i. $w_{ji}$ is weight of link that connect node j and node i. If a network is unweighted, degree is the number of neighbor nodes.

Second, calculate the adjacency degree ($A_i$) of each node.

$$A_i = \sum_{j \in \Gamma_i} k_j \qquad\qquad - \quad (3)$$

$A_i$ is total weight of neighbor nodes of node i.

10) Comments about Multiresolution detection method : does the path in Line 41 mean the shortest path ?

➤ The path in calculating link intensity is 'Distinct path'. k-distinct path can be denotead path between node i and j with k-edges if the path has no identical intermediate nodes or edges with any other distinct paths. We added this explain into 3.3.

- The First, Define Distinct path

The simple(not repeating links between nodes) and elementary (not repeating nodes) path θ between node i and j with k-edges is denoted as a k-edge distinct path, if the path has no identical intermediate nodes or edges with any other distinct paths.

Second, calculate the link intensity ($I_P$) of each link.

$$I_P(e_{ij}) = \begin{cases} \sum_{p=1}^{P} \alpha_p \times \dfrac{\sigma(path_p(v_i, v_j))}{\min(w_i, w_j)}, & e_{ij} \in E \\ 0, & otherwise \end{cases} \qquad - \quad (6)$$

Here, $\sigma(path_p(v_i, v_j))$ is the sum of link weights in p-edge distinct paths from node i ($v_i$) to node j ($v_j$). P is the parameter of the path, and $\alpha_p$ is a polygonal effect parameter. For edge $e_{ij}$ between node i and node j, $w_i$ and $w_j$ are their respective strengths.

Third, identify the links with link intensity greater than the selected threshold and create a group of nodes with the identified links.

$$\begin{aligned} v_j \in V, & \qquad I_P(e_{ij}) > t \\ v_j \in c_u, & \qquad I_P(e_{ij}) > t \end{aligned} \qquad\qquad - \quad (7)$$

Here, t ($0 < t \le 1$) is the selected threshold, and $c_u$ is a group of nodes.

Third, calculate the belonging coefficient ($I_P$) of the nodes in node set u ($c_u$).

$$I_P(c_u, v_j) = \sum_{v_i \in c_u} I_P(e_{ij}) \qquad\qquad - \quad (8)$$

This method has the advantages of forming groups more accurately and faster than other methods.

11) Comments about Table 2 : Suggest the table should be able to fit the whole table on one page instead of overflow.

➢ We changed the table's width and put Talbe 2 on one page.

**Table 2**. Average, maximum, and minimum link weights of each node; Parentheses next to link weights are nodes that form a maximum or minimum value for target nodes;

| Node | Average | Maximum (Node) | Minimum (Node) |
|---|---|---|---|
| Pearl River Delta | 0.352 | 1.674 (Hong Kong SAR) | 0.203 (Tokyo) |
| Tokyo | 0.226 | 0.528 (Nagoya) | 0.140 (Tianjin) |
| Shanghai | 0.253 | 1.076 (Hangzhou) | 0.153 (Tokyo) |
| Beijing | 0.232 | 0.850 (Tianjin) | 0.130 (Osaka) |
| Manila | 0.294 | 0.467 (Ho Chi Minh City) | 0.219 (Shanghai) |
| Seoul | 0.227 | 0.276 (Ha Noi) | 0.155 (Tokyo) |
| Osaka | 0.244 | 0.870 (Nagoya) | 0.130 (Beijing) |
| Bangkok | 0.272 | 0.520 (Ho Chi Minh City) | 0.194 (Shanghai) |
| Tianjin | 0.252 | 0.850 (Beijing) | 0.143 (Osaka) |
| Shantou | 0.284 | 0.861 (Hong Kong | 0.183 (Beijing) |
| Chengdu | 0.293 | 0.621 (Chongqing) | 0.200 (Shanghai) |
| Ho Chi Minh City | 0.308 | 0.520 (Bangkok) | 0.217 (Shanghai) |
| Nagoya | 0.254 | 0.870 (Osaka) | 0.139 (Beijing) |
| Wuhan | 0.292 | 0.529 (Hangzhou) | 0.195 (Taipei City) |
| Hong Kong SAR | 0.364 | 1.674 (Pearl River Delta) | 0.215 (Tokyo) |
| Shenyang | 0.240 | 0.367 (Tianjin) | 0.166 (Tokyo) |
| Taipei | 0.220 | 0.333 (Shantou) | 0.147 (Beijing) |
| Hangzhou | 0.279 | 1.076 (Shanghai) | 0.164 (Beijing) |
| Kuala Lumpur | 0.308 | 0.377 (Wuhan) | 0.207 (Beijing) |
| Xi'an | 0.271 | 0.508 (Chengdu) | 0.188 (Osaka) |
| Ha Noi | 0.341 | 1.162 (Hai Phong) | 0.238 (Tokyo) |
| Chongqing | 0.289 | 0.621 (Chengdu) | 0.207 (Beijing) |
| Cebu | 0.246 | 0.354 (Manila) | 0.179 (Beijing) |
| Hai Phong | 0.342 | 1.162 (Ha Noi) | 0.235 (Shanghai) |

12) About mutual information results, certain cities have the smallest value with other cities. In the paper, author told that this result came from the location of cities. However, the reason for the result is judged to be due to climate, not location. Therefore, authoer need to elaborate about it.

➢ When looking at the cities in the Minimum column of Table 2, except for Taipei, cities are located in the outskirt of the study area. Regarding their location, we initially hypothesized that their locations would cause different rainfall patterns from cities in central, which makes low value in mutual information. However, after receiving the above comment, we agreed that another potential factor might be climatic. We will mention that this is one of the limitations of our study and that future studies can explore this. We propose that future studies analyze each city's rainfall characteristic with geographical data and compare them to find a mechanism behind these different rainfall patterns. In the revised version, we changed Line 162 related to the comment like below and put this problem in the discussion part.

- (Line 162) Two cities have a common feature that their location is in the outskirt of the study area.
- (Discussion) In the mutual information result, we could find that many cities had the lowest value with Tokyo and Beijing. We tried to find why this result came out, but we could not find any differences in the rainfall data. Therefore, future studies should collect and analyze other climate factors and geographical factors for finding unique rainfall characteristics in Tokyo and Beijing.

13) It would be helpful put the reference first line in 4.2 section.

➢ Thanks for your comment about putting reference into the first line of 4.2 section. However, we already put the reference, which the reviewer recommended to us, in 3.2 section (3.2 Vital node identification using adjacency information entropy). In section 3.2, readers can recognize which paper has background and detailed information about the vital node identification method.

14) Is "95$^{th}$ quartile" right express? Quartiles are 25%, 50% and 75%.

➢ We did not correctly express the word "95$^{th}$ quartile". Quantiles are points in a distribution related to the rand order of values in that distribution. On the other hand, quartile refers to the point where the data is divided into quarters, as the reviewer commented. Therefore, we accordingly fixed this, and now we express "95$^{th}$ quantile".

- (Line 179) The threshold value is the 95th quantile of the calculated belonging coefficient, 0.06, in order to form a group of nodes with strong relationship.

15) Where is location of sea of Okhotsk?

➢ The sea of Okhotsk is marginal sea of the western Pacific ocean. It is located between Russia's Kamchatka penisula on the east, the Kuril islands on the southeast, Japan's island of Hokkaido on the sotuh, the island of Sakhalin along the west, and a stretch of eastern Siberian coast along the west and north. We tried to express the location of the sea of Oktotsk in the figures but it is far from study area. Therefore we added the location in Line 211.

[Figure]

**< Location of Sea of Okhotsk (from Google map) >**

- (Line 211) The Indian monsoon moves northwest from the Bay of Bengal, passing mainland China into the Sea of Okhotsk, which is located between the Russia's Kamchatka peninsula and Japan's island of Sakhalin.

16) Comment on Reference: Please separate each article, and/or ident the lines after the lead one. This will make it easier for the reader to find one referred to in the text.

➢ Now we clearly understood and revised all list of references accordingly.

17) Comment on English expression

➢ We checked all the comments related to English expression. We appreciate comments on English expression. Once again, we checked the English and appropriately revised the whole manuscript. These changes can be seen with the track-change version of our revised paper.

---

## Author Response (AR3)

**Author's response**

We got third review from three reviewers during discussion interval. The reviewers talked about many factors in the paper. We tried to reflect comments on revised manuscript. We believe that the paper is substantially improved as the result of the revision. We wrote point-by-point response (in blue) to reviewer's comments (in red). Also, we put revised part of the manuscript in black.

**\<Reviewer 1\>**

1)  (Line 24) ″occurred worldwide during 2000 to 2020″, it should be: ″during the period from 2000 to 2020″.

> ➢ Following the above comment, we changed the Line 24 like below.

- (Line 24) an annual average of 165 flood disasters occurred worldwide during the period from 2000 to 2020……

2)  (Line 65) ″Using the correlation coefficient makes that a network reflect relationship between region″: this pharse should be rewritten in a better way

> ➢ The current expression about Line 62 is not clear for understanding the meaning. Therefore we changed it like below.

- (Line 65) By using the correlation coefficients as weights, it makes the network to reflect relationships between regions.

**<Reviewer 3>**

**1. Major revisions**

1) The authors rank the importance of the nodes (cities) based on the value of adjacency information entropy and average link weight. They have explained why some locations have a low rank. However, there is no physical interpretations of the cities having very high/low entropy values. Are these cities more prone to flooding due to extreme rainfall from monsoon/typhoons from the South China Sea ? Also do they know any reason why the group of cities in the north of the map have low values compared to those south ? Such an explanation would explain the purpose of conducting the vital node identifictation methodology.

➢ In vital node identification analysis, nodes with high ranks have many links with other nodes and their link weights have large value. This study constructed an East Asia rainfall network using similarity of rainfall between 24 major cities. Therefore if a node has high similarity with other nodes, it would contain many links having large weights and place at a high rank in the vital node identification analysis. The nodes in high rank have a common thing that they are located near the major moisture source of East Asia (Hong Kong SAR, Pearl River Delta, Shantou – The South China Sea; Kuala Lumpur and Ho Chi Minh city – The Indian monsoon ; Manila – The East Asian monsoon). We checked the rainfall of the high-rank cities and they have higher rainfall intensity and rainfall days than other cities. On the other hand, low-rank cities are commonly located in the north part of study area have lower rainfall intensity and rainfall days because they are far from the moisture sources. Through the vital node identification, we could check the major moisture sources in East Aisa. We added the results of the vital node identification in parts 4.2 and 4.5. We hope readers understand what we found in the vital node identification analysis.

- (Line 195-197) We also find another common thing between high-rank cities that they are located in the beginning of impact of the main influence factors in East Asia rainfall. We will describe this in Section 4.5.
- (Line 271-285) The adjacency information entropy was calculated and compared to check the effects of nodes in the network. The results indicate that nodes surrounding the South China Sea and node at the beginning of two monsoons (Indian and East Asian) were highly ranked, and node's location is one of the essential factors in identifying vital nodes. In the rainfall complex network research, the high rank nodes mean that they are important sites for the propagation of rainfall event. Based on the interpretation of high-rank node, we verified that the South China Sea and two monsoons are the primary moisture sources in East Asia. The South China Sea supplies a huge amount of moisture into East Asia, and the two monsoons pass through it. Vapour from the South China Sea first affects coastal cities and then moves to other cities in the continent. Thus, rainfall from some cities affects the neighbouring cities. Based on this phenomenon, cities in the South China Sea ranked high. Kuala Lumpur and Manila are also in high ranks. They have a

common thing their location is at the beginning of each major monsoon influence. The two monsoons pass Kuala Lumpur and Manila first and then move like Figure 7. Because of these characteristics, high-rank nodes have higher rainfall intensity and the number of rainfall days compared to other cities. On the other hand, low-rank cities have low rainfall intensity and the rainfall days. Because the low-rank cities commonly are located in the north which is very far from the moisture sources, the low rank cities get less moisture and have less similarity to other cities. Through vital node identification, we could find the major rainfall influence elements in East Asia and it help to interpret the relationship between groups in Section 4.4.

2) I like how the clustering analysis identify the cities having similar rainfall patterns. However, the authors only explain about Seoul and Kuala Lumpur as to why they are isolated from all groups. There is no interpretation of the cities with form groups. I have a feeling that some groups may be formed because the particular cities likes in the basin of the same river, for instance, G4 is in the Yellow river basin and G3 in the middle/lower reaches of the Yangtze, which maybe the cause of similar rainfall characteristics. Such knowledge is important to predict concurrent floods in multiple locations. Could you please check if such is the case of most clusters ?

➢ To check the reviewer's comment, we used global basin and river data from HydroSHEDs. Through the data, we found that nodes of some groups are located in the same basin or have the same river. However, not all groups had the above characteristic. The multiresolution community detection is based on link weight. As mentioned in the answer about the first comment, because the network is based on rainfall similarity, we could judge that nodes forming the same group have high rainfall similarities with each other, and also similar rainfall characteristics. The rainfall is caused by a combination of various factors such as weather, topographic and hydrologic elements. Therefore, if two regions have a high similarity, we could interpret that the regions have many same common factors which are affecting rainfall. However, since this study used only rainfall data, we could not confirm which factors make nodes have high rainfall similarity. These factors are very important to finding rainfall relationships and also predict concurrent floods in multiple locations as the reviewer mentioned. Therefore, in the future study, we will focus on this part and find the common factors among the groups.

- (Line 289-295) We also tried to find common physical factors between nodes in the same group. We found that nodes of some groups are located in the same basin or share the river. However, these factors do not apply to all groups. The cluster analysis result is conducted based on the similarity of rainfall. Rainfall is a meteorological phenomenon caused by a combination of various factors such as geographic, hydrologic, meteorological and ecological elements. If two regions have high similarity in rainfall, they share similar characteristics of factors influencing rainfall. The study tried to find common factors that make groups, but

failed to find them. These factors are essential to predict concurrent floods in multiple locations. Therefore, in the future study, we will try to find what factors make the high similarity between nodes in the same group by using geographic, meteorological, and hydrological data.

3) I wonder if lowering the threshold value in the clustering analysis, leads to bigger clusters and unravels grouping together of those clusters which are found to be related in the time lag analysis. For instance, G2, G3, G4 and G6 may be related as they are linked by the anticyclone and two ore more the groups fall in the same cluster when a lower threshold is chosen. Has the authros verified that ?

➢ We conducted cluster analysis according to threshod by 2.5% intervals from 95% to 75%. In the cluster analysis result, When thresold became 92.5%, Group 5 and Group 7 were merged. After 85%, Group 3 and Goup 6 became the same group. In 80% and 77.5%, Group 8 was added to Group 5+7 and then Kuala Lumpur add to Group 5+7+8. Rather than what the reviewer expected, we confirmed that the groups in Southwest were firstly being merged. And then there is a merger in the middle part of the study area. In the maximum cross mutual information result, groups at southwest part had a higher value between them compared to groups in the north. This means that southwest groups have a high simlarity of rainfall to each other. Therefore, they become a huge cluster when the threshold is 75%. We added the result of cluster analysis according to the various threshold in part 4.5.

- (Line 298-302) This result could also be confirmed when creating groups using various thresholds in Section 4.3. When threshold became 92.5%, Group 5 and Group 7 were merged. After 85%, Group 3 and Group6 became same group. In 80% and 77.5%, Group 8 was added to Group 5+7 and then Kuala Lumpur add to Group 5+7+8. As you can see in Figure 6. Group 7 and Group 8 has strong relationship with Group 5 because of the two monsoon and Group 3 and Group 6 also have high relationship because of anomalous anticyclones.

-

4) Since the authors used the rainfall time series of the whole year for the anlaysis, how do they know while climatologically imerpreting in Sec. 4.4, that the relation between groups is due to Indian monsoon which takes place in summer, or that the anticyclone is not specific to a certain season ?

➢ In East Asia, 80-90% of rainfall occurs in summer due to monsoon and typhoons in summer. Therefore, most of the characteristics of rainfall are formed by summer phenomena. Because of characteristics of rainfall in East Asia, past studies have also conducted research on summer phenomena to find relationships, features and weather system of rainfall. Accordingly, this study also conducted analyzing rainfall relationships based on summer phenomena. We added the above reason in part 4.4.

- (Line 239-240) For founding reasons of rainfall relationship in East Asia, it need to analyze East Asian summer rainfall system. Because East Asian summer rainfall

contain more 90% of total rainfall in East Asia (Chen et al., 2015). The relationships in figure 6 are derived from synoptic atmospheric circulation in East Asia. Indian and East Asian monsoons are major factors affecting rainfall in East Asia (Chen et al., 2015).

- (References) Chen, F., Xu, Q., Chen, J., Briks, H. J. B., Liu, J., Zhang, S., Jin, S., An, C., Telfod, R. J., Cao, X., Selvaraj, K., Lu, H., Li, Y., Zheng, Z., Wang, H., Zhou, A., Dong, G., Zhang, J., Huang, X., Bloemendal, J., and Rao, Z.: East Asian summer monsoon precipitation variability since the last deglaciation, Scientific Reports, 5, 1186, https://doi.org/10.1038/srep11186, 2015.

**2. Minor revisions**

1) The main conclusions of the paper is not clear when reading the abstract. The concluding statement should talk more about the authors' findings.

➢ We agree with the comment about the abstract. We expressed the result of the study very briefly. Therefore, this time, we added more detailed results of the study in the abstract. Through the revised abstract, readers will be able to better understand the contents and results of the paper.

- (Abstract) Concurrent floods in multiple locations pose systemic risks to the interconnected economy in East Asia through supply chains. Despite the significant economic impacts, the understanding of the interconnection between rainfall patterns in the region is yet limited. Here, we analyzed spatial dependence in rainfall patterns of the 24 mega-cities in the region using complex analysis theory and discussed the technique's applicability. Each city and rainfall similarity was represented by a node and a link, respectively. Vital node identification and clustering analysis were conducted using adjacency information entropy and multi-community detection. In the vital node identification analysis result, high-rank nodes are cities that are located near main vapor providers in East Asia. Through the multi-community detection, the groups were clustered to reflect the spatial characteristics of the climate. In addition, the climate links between each group were identified through the cross-mutual information considering the delay time for each group. We found a strong bond between northeast China and the south Indochina Peninsula and verified that the links between each group originated from the summer climate characteristics of East Asia. The result of the study shows that complex network analysis could be a valuable method for analyzing the spatial relationship between climate factors.

2) There is a consistent confusion created by the consistent usage of the term "correlation coefficient" synonymously in place of "similarity measure". Since correlation coefficient is a type of linear simillartiy measure, whereas mutual information is nonlinear, they are different. It is also not clear in the abstract, introduction and methodology Sec. 3.1.

Therefore whether the authors use mutual information or correlation in their work, Kindly correct it.

➢ We understood "Correlation coefficient" as a word containing both linear and non-linear correlations. After checking the comment about the correlation coefficient, we investigated it and found that the real meaning is not the same as what we knew. Therefore, we changed all the terms related to the correlation coefficient in our paper. Belows are some parts of revised content in the paper. Please check the whole revised thing in the paper with track version.

- (Line 66-70) While no perfect methodology exists to clearly address this challenge, new methodologies are constantly being proposed. In this study, we assumed that each region (node) is connected with all the other regions (nodes) in the network, and that each connection (link) has a similarity as a weight. By using the similarity measures as weights, it makes the network reflect relationships between regions. Using the similarity measure makes a network reflect the relationship between region.
- (Line 121-123) The most widely used methodology is the similarity measure (Donges et al., 2009). Depending on the value of the similarity calculated between two nodes, the researcher can define whether a link exists.
- (Line 129-130) MI can consider the nonlinearity of the data and has the advantage of calculating the similarity between different data sizes (Goyal, 2014).

3) Figure 5 has a typo in labelling of boxes G2 and G3.

➢ We checked Figure 5 and revised the Figure 5 like below.

[Figure]

**Figure 1**. Group of nodes using multiresolution community detection; there are 8 groups in the

East Asia; G1(Pearl River Delta, Hong Kong SAR, Shantou, Taipei City), G2(Osaka, Nagoya, Tokyo), G3(Wuhan, Hangzhou, Shanghai), G4(Tianjin, Shenyang, Beijing), G5(Bangkok, Ho Chi Minh City), G6(Xi'an, Chengdu, Chongqing), G7(Hanoi, Haiphong), G8(Manila, Cebu); Seoul and Kuala Lumpur did not make group with other nodes.

4) In the data, it is not clear if you use a single grid point from APRODITE to represent each city of a group of grid points.

   ➢ In this paper, the grids where each city belongs were identified from the APRODITE data which have 0.25x0.25 degree size. And rainfall data was extracted from the grid for constructing rainfall data of 24 major cities in East Asia.

      - (Line 94-95) We extracted the rainfall data from the grid($0.25° \times 0.25°$) where each city belongs.

5) Line 58-59 in the introduction is incorrect. A complex network may be of several types. What the authors seems to refer to are functional climate networks in which connectivity is defined on the basis of "statistical interdependence" between the pairs of nodes. Please change the sentence.

   ➢ In the cause of complex networks in hydrology and meteorology fields, most of them used the statistical interdependence method. We generalized and expressed the above sentence in Line 58-59. However, there are various link definition methods for making complex network according to the subject. Therefore, the sentence was modified by making the expression more clearly as a complex network in the hydrology and meteorology fields.

      - (Line 61-62) The complex networks in hydrology and meteorology fields defined connectivity using statistical interdependence methods.

6) Line 106 in Sec 3.1, a node need not be a fixed element. A node in a complex network is a dynamical unit, it may be fixed, movable, disappearing. In case of climate network, the nodes are spatial grid points which represent a dynamical subsystem. Kindely correct the sentence.

   ➢ Line 106 is also thought to be the same problem with comments 5 in minior revisions. We revised Line 106 as the most common definition of a node. Also, we add a reference to the paper which contains the definition of a node.

      - (Line 111-112) A node represents some entity or agent that serves as a point of intersection/junction within a network (Kivelä et al., 2014).

- (Reference) Kivelä, M., Arenas, A., Barthelemy, M., Gleeson, J. P., Moreno, Y., and Porter, M. A.: Multilayer networks, Journal of Complex Networks, 2(3), 203-271, https://doi.org/10.1093/comnet/cnu016, 2014.

7) Sec. 3.2 in methodology, please add a sentence on the relevance of vital nodes or high degree nodes in climate system.

➢ We wrote down the meaning of important nodes in rainfall complex networks by refering to related studies. Vital nodes in the rainfall complex network are important points for the propagation of rainfall event.

- (Line 146) In the rainfall studies, vital nodes are interpreted as important points for propagation of rainfall event.

8) Sentence "Each node has different link weights" on line 169 is confusing, as link weight is between pairs of nodes. Please clarify or remove.

➢ Line 169 makes readers confusing. So, we deleted Line 169.

9) In line 175, authors talk about narrower or wider range of average, maximum and minimum link weights. They can use standard deviation of the range of av., max., min. Link weights instread of quantify the "range".

➢ We agree that it is better to express range as standard deviation. We calculated the standard deviation of average, minimum and maximum vales and then compared them.

- (Line 181-182) According to Table 2, the ranges of average, maximum, and minimum link weights are 0.22–0.37, 0.27–1.67, and 0.13–0.24, respectively. The standard deviation of average, minimum and maximum values are 0.041, 0.033 and 0.394. The standard deviation of maximum values had 10 times larger value than the minimum values.

10) Section 4.2, please specify that high ranking nodes means node of high adjacency.

➢ To make the readers know about the result meaning, we clarified the Line 183-184 like below.

- (Line 190) The cities with high ranking nodes are located around the South China Sea (Fig. 4). In addition, they have a high adjacency in the adjacency matrix.

11) I do not understand in lines 186-187. 'High ranking nodes in center of the maps.... low-ranking nodes are diametrically opposed.... ' Kuala, a high ranked node is not in

center. ′diametrically opposite′ is not clear. Please remove the explanation if it is not needed.

➢ Line 186-187 had been added in the Second revision because another reviewer asked for it. But we agree that it is not clear to readers. We eliminated the Line 186-187.

12) Line 184-185 ′low mean of MI′ and ′large average of MI′, do you refer to the second column of average link weight in Table 2 ? Please clarity.

➢ Both are meaning of the average value of mutual information in Table 2. We changed 184-185 for clarifying the meaning.

- (Line 190-191) Cities with low-rank nodes are in the northeast outskirts, except for Taipei, and have a low value of average MI.

13) Line 233 onwards, Indian monsoon does not move northwest, rather the moisture bearing cross equatorial south-westerly low level jest coming from the Somalian coast, which causes Indian monsoon, moves northwest.

➢ One of the important thing generating rainfall events is supply and moving of moisture. We interpreted that moisture from the Indian monsoon is moving to the northwest direction by the monsoon. However, after receiving the above comment, we searched about the related information and checked many related papers. We finally conclude that low-level jest move moisture to East Asia region. Therefore, we revised all related sentences after Line 233.

- (Line 246-249) Water vapor from Indian monsoon moves northwest from the Bay of Bengal, passing mainland China into the Sea of Okhotsk, which is located between the Russia's Kamchatka peninsula and Japan's island of Sakhalin. G5, G6, and G7 in this pathway are related to each other by the Indian monsoon. The movement of water vapor from Indian monsoon is caused by low level jet stream from the Somalian coast.

14) Replace ′vapour provider′ and ′vapour′ in line 258 with ′moisture source′ and ′water vapour′ respectively.

➢ We revised "Vapour provider" as "moisture source".

- (Line 274-275) Based on the interpretation of high-rank node, we verified that the South China Sea and two monsoons are the primary moisture sources in East Asia.

15) Mention 'belonging coefficient' specifically in line 261.

➢ We revied "The coefficient" as "the belonging coefficent".

- (Line 286) As described in Section 4.3, the belonging coefficient of each node was calculated by using the link weight.

**<Reviewer 4>**

1) Why not directly use the original precipitation points (or maybe a slight coarsen resolution, if the orginal resolution is too small), because i see from Fig. 2 that in East Asian region, there are a lot of points more than 24 ? 24-city is a quite small-scale. That means the constructed networks has only 24 nodes. Even for the selected 24 cities, there are probably several stations insdie each of them. Then it would be better to consider the original dataset, because the aggregation of rainfall data for each city is essentially omitting some local information. Besides, as far as i know, the computational complexity (that is, the actual running time) for a network with fro example 1000 nodes, can still be pretty small. Another reason to consider a higher resolution is that is delivers a more convincing conclusion ?

   ➢ This study is one of the following studies about "Correlated Risk for Heavy Precipitation in Mega-cities in East Asia" in references. In "Correlated Risk for Heavy Precipitation in Mega-cities in East Asia" study, we analyzed the relationship between several weather-ocean events (El Nino and Southern Oscillation, West Pacific pattern, Idian Ocean Dipole etcs) and rainfall of major cities in East Asia. In the current study, we focused on the relationship between the major cities and evaluation of complex network analysis methods in weather system research. Through all related studies, we intend to finally analyze the rainfall of the entire East Asia region by applying a complex network methodology.
   We would like to apply the complex network method to a small scale network first. Because we could easily understand and interpret the subject before applying to a huge scale network. Therefore, we only treated the major cities of East Asia in this study. In the future study, we will make a rainfall network of East Asia consisting of more than 1,000 nodes and analyze the network with various analysis method. Regarding rainfall data, we needed grid data for conducting the final research. In the beginning of the study, we evaluated several grid data and selected the APHRODITE data which provides high-resolution rainfall data about East Asia. However, since the APHRODITE data has several problems which the reviewer mentioned in the comment, we will compare the APHRODITE data with observed data for checking missing data in the future study.

2) Why use vital node identification and community detection together ? And how they both contribute to the final conclusion ? In the domain of complex network analysis, vital node identification and community detection are different research probelms, although there might be some overlapping between then. I initially thought there might be some connections between the results obtained from teses two methods in this manuscript. But it seems to me that they are separate contents. In my opinion, vital nodes are possbily useful for disaster mitigation, while community detection could be useful for finding out regions of similar climatological behavior, but there might be a lack of explanation on why they should be considered together, and specifically, how they help in obtaining the final conclusion.

➢ The study used vital node identification and multiresolution community detection method. Vital node identification method identifies important nodes in a network. In the rainfall network, high-rank nodes mean important places for the propagation of rainfall. In the vital node identification result, nodes in high rank have a common thing that they are located near the major moisture source of East Asia (Hong Kong SAR, Pearl River Delta, Shantou – The South China Sea; Kuala Lumpur and Ho Chi Minh city – The Indian monsoon ; Manila – The East Asian monsoon). We checked the rainfall of the high-rank cities, and they have higher rainfall intensity and rainfall days than other cities. On the other hand, low-rank cities are commonly located in the north part of the study area have lower rainfall intensity and rainfall days because they are far from the moisture sources. Multiresolution community detection method can cluster nodes with similar characteristic in a network according to the threshold. In the study, cities with similar rainfall characteristics were clustered through community detection, and the network could be simplified through clustering. The two methodologies had different results, but the conclusions have helped interpret rainfall relationship between major cities in East Asia. Through the vital node identification, we could check the major moisture sources in East Asia and help to interpret the relationship between groups. We added what we got from both methods and how they helped to interpret the rainfall system in part 4.5.

- (Line 310-317) During the analysis, vital node identification helped to identify important sites for propagation of rainfall and major moisture sources in East Asia. The vital node identification is a useful method to analysis the system or phenomena. Therefore, it can be used for researching natural disaster or meteorological system. Multiresolution community detection method found cities having similar rainfall characteristics according to threshold. It made the groups having similar characteristics and simplified the rainfall network. This helped to understand rainfall relationship between the East Asia major cities. Two methods draw out different results but, they help to interpret the results. For example, the major moisture sources found by vital identification helped to explain the relationship between groups. In addition, their results ultimately helped to understand the rainfall system in East Asia.

3) Regarding the community detection, how is the quality of community structure (or groups in this manuscript) measured? Also, how is the threshold ″t″ determined? When applying community detection, commonly accpeted measures include modularity and normalized mutual information. But there is a lack of explanation in this regard. The threshold ″t″ could be any value between the range of [0,1]. Different values could lead to different forms of groups. If there is not a standard way to determine what it would be, then why the group in Fig. 5 would be chosen as the final result.

➢ The study evaluated the results of clustering with Newman-Girvan modularity. We set a threshold group divided by 2.5% interval from 95% to 75% and applied the group to

the network. We calculated Newman-Girvan modularity about all cluster results and found that 95% threshold had the highest value in modularity. In Newman-Girvan modularity, a clustering result can get high scores if they are densely connected internally, but only weakly connected to other group. Based on the above result, we concluded that the 95% threshold is the best parameter for clustering analysis.

For selecting threshold problem, there is no clear method. The developer of the multiresolution community detection method also pointed out the threshold selecting problem in their paper. However, they showed that the larger the threshold, the more clusters and more details can be heard. Therefore, the study performed cluster analysis by applying a high threshold. We added the cluster evaluation methodology and its result in parts 3.3 and 4.3 each.

- (Line 166-171) For evaluating the result of the cluster method, we used Newman-Girvan modularity. The Newman-Girvan modularity method compares the number of links connecting nodes inside a group of nodes with an expectation of this number under a random null model (Newman & Girvan, 2004). The modularity (Q) is calculated by Eq. (9). Where, P is a cluster of node groups ($P = \{g_1, g_2, \cdots, g_a\}$), and $g_a$ is a node group. m is a total weight of links. The modularity measure assigns high scores to communities if they are densely connected internally, but only weakly connected to other group. We set threshold groups divided by 2.5% intervals from 95% to 75% and calculated the Newman-Girvan modularity according to the threshold groups.

$$Q = \frac{1}{2m} \sum_{P \in g_a} \sum_{i,j \in g_a} (A_{ij} - \frac{k_i k_j}{2m})$$
- (9)

- (Line 216-218) For evaluating the cluster result, we calculated the Newman-Grivan modularity. We set the divided by 2.5% intervals from 95% to 75%. In the modularity result, 95% show the largest modularity (Table 3). In the Table 3, there are thresholds having same value of modularity. Those thresholds have same result of clustering.

**Table 1**. Result of Newman-Grivan modularity according to threshold

| Threshold | 95% | 92.5% | 90% | 87.5% | 85% | 82.5% | 80% | 77.5% | 75% |
|---|---|---|---|---|---|---|---|---|---|
| Modularity | 0.0313 | 0.0311 | 0.0311 | 0.0311 | 0.0309 | 0.0309 | 0.0294 | 0.0286 | 0.0286 |

4) From the current version of the Abstract, the main conclusion is not emphasized

➤ We agree with the comment about the abstract. We expressed the result of the study very briefly. Therefore, this time, we added more detailed result of the study in the abstract. Through the revised abstract, readers will be able to better understand the contents and results of the paper.

- (Abstract) Concurrent floods in multiple locations pose systemic risks to the interconnected economy in East Asia through supply chains. Despite the significant

economic impacts, the understanding of the interconnection between rainfall patterns in the region is yet limited. Here, we analyzed spatial dependence in rainfall patterns of the 24 mega-cities in the region using complex analysis theory and discussed the technique's applicability. Each city and rainfall similarity was represented by a node and a link, respectively. Vital node identification and clustering analysis were conducted using adjacency information entropy and multi-community detection. In the vital node identification analysis result, high-rank nodes are cities located near main vapor providers in East Asia. Through the multi-community detection, the groups were clustered to reflect the spatial characteristics of the climate. In addition, the climate links between each group were identified through the cross-mutual information considering the delay time for each group. We found a strong bond between northeast China and the south Indochina Peninsula and verified that the links between each group originated from the summer climate characteristics of East Asia. The result of the study show that complex network analysis could be a valuable method for analyzing the spatial relationship between climate factors.

5) Contents between 153 and 165 (both equations and paragraphs) should be reorganized in a more fomral way.

➢ We checked HESS instruction and recently published papers in HESS journal and then edited not only Lines 153-165 but also Lines 135-144.

- (Line 140-146) The first step is calculating strength of each node in a weighted network (Eq. (2)).

$$k_i = \sum_{j \in \Gamma_i} w_{ji} \qquad \text{-} \quad (2)$$

where, j is the neighbor of node i. $\Gamma_i$ is the set of neighbors of node i. $w_{ji}$ is weight of link that connect node j and node i. If a network is unweighted, degree is the number of neighbor nodes. Next, estimate an adjacency degree of each node (Eq. (3)).

$$A_i = \sum_{j \in \Gamma_i} k_j \qquad \text{-} \quad (3)$$

$A_i$ is an adjacency degree which means total weight of neighbor nodes of node i. Based on Eq. (2) and Eq. (3), selection probability can be calculated (Eq. (4)). Eventually an adjacency information entropy of a node is calculated based on Eq. (5).

$$P_{i_j} = k_i / A_j \qquad \text{-} \quad (4)$$

$$E_i = \sum_{j \in \Gamma_j} (P_{i_j} log_2 P_{i_j}) \qquad \text{-} \quad (5)$$

After comparing the calculated adjacency information entropy of each node, the importance is determined according to the descending power. In the rainfall studies, vital nodes are interpreted as important points for propagation of rainfall event.

- (Line 154-166) This method has the advantages of forming groups more accurately and faster than other methods. For making groups, belonging coefficient of the nodes should be calculated. First step for estimating belonging coefficient is defining a distinct path. The simple (not repeating links between nodes) and elementary (not repeating nodes) path θ between node i and j with k-edges is denoted as a k-edge distinct path, if the path has no identical intermediate nodes or edges with any other distinct paths. After defining distinct paths, it need to calculated link intensity of each link. The equation of link intensity is same as Eq. (6).

$$I_P(e_{ij}) = \begin{cases} \sum_{p=1}^{P} \alpha_p \times \dfrac{\sigma(path_p(v_i, v_j))}{\min(w_i, w_j)}, & e_{ij} \in E \\ 0, & otherwise \end{cases} \qquad - \quad (6)$$

Where, $\sigma(path_p(v_i, v_j))$ is the sum of link weights in p-edge distinct paths from node i $(v_i)$ to node j $(v_j)$. P is the parameter of the path, and $\alpha_p$ is a polygonal effect parameter. For edge $e_{ij}$ between node i and node j, $w_i$ and $w_j$ are their respective strengths. Based on link intensity, find links which have larger value of link intensity than a threshold and create a group of nodes with the identified links (Eq. (7)).

$$v_j = \begin{cases} v_j \in V, & I_P(e_{ij}) > t \\ v_j \in c_u, & I_P(e_{ij}) > t \end{cases} \qquad - \quad (7)$$

Where, t $(0 < t \le 1)$ is the selected threshold, and $c_u$ is a group of nodes. The threshold is determined according to researcher personal view. Final step is calculating belonging coefficient $(I_P)$ of the nodes in node set u $(c_u)$ (Eq. (8)).

$$I_P(c_u, v_j) = \sum_{v_i \in c_u} I_P(e_{ij}) \qquad - \quad (8)$$

6) Fig. 5 has a label probel, two "G2".

➢ We checked Figure 5 and revised the Figure 5 like below.

[Figure]

**Figure 2**. Group of nodes using multiresolution community detection; there are 8 groups in the East Asia; G1(Pearl River Delta, Hong Kong SAR, Shantou, Taipei City), G2(Osaka, Nagoya, Tokyo), G3(Wuhan, Hangzhou, Shanghai), G4(Tianjin, Shenyang, Beijing), G5(Bangkok, Ho Chi Minh City), G6(Xi'an, Chengdu, Chongqing), G7(Hanoi, Haiphong), G8(Manila, Cebu); Seoul and Kuala Lumpur did not make group with other nodes.

---

## Author Response (AR4)

**Author's response**

We got fourth review from two reviewers during discussion interval. The reviewers talked about many factors in the paper. We tried to reflect comments on revised manuscript. We believe that the paper is substantially improved as the result of the revision. We wrote point-by-point response (in blue) to reviewer's comments (in red). Also, we put revised part of the manuscript in black.

**\<Reviewer 4\>**

1)  Please check if is reasonable to use the temr 'Multi-community detection'. Originally, it should be 'Multiresolution community detection'. I don't think this abbreviation delivers the same meaning.

    ➢ We checked meaning of the both words and agreed with the reviwer opinion. So, we changed the term as 'Multiresolution communtiy detection' in abstract, application and results and conclusion.

    - (Line 15) ...... entropy and multiresolution community detection.
    - (Line 22) Keywords: ......., Multiresolution community detection
    - (Line 150) Therefore, the multiresolution community detection method can ....
    - (Line 322) .... Vital node identification and multiresolution community detection ...

2)  Plase change Eqs. 6 and 7 in a more formal way. Should not there be parentheses if they are condition al formulas ?

    ➢ We checked Eqs. 6 and 7 and then changed the equations like bleow.

    - $$I_P(e_{ij}) = \begin{cases} \sum_{p=1}^{P} \alpha_p \times \frac{\sigma(path_p(v_i,v_j))}{\min(w_i,w_j)} & , \quad e_{ij} \in E \\ 0, & \quad otherwise \end{cases} \tag{6}$$

    - $$v_j = \begin{cases} v_j \in V, \ I_P(e_{ij}) > t \\ v_j \in c_u, \ I_P(e_{ij}) > t \end{cases} \tag{7}$$

3)  Please check Eq. 9, make sure it is in the right form. If i remember correctly, the current form is not correct. Should be g(a) ∈ P, isn't it ?

    ➢ We checked Eq. 9 and found that expression of the first sigma. We changed the Eq. 9 like below

    - $$Q = \frac{1}{2m} \sum_{g_a \in P} \sum_{i,j \in g_a} (A_{ij} - \frac{k_i k_j}{2m}) \tag{9}$$